# Implicit Variational Inference
# for High-Dimensional Posteriors

**Anshuk Uppal**
Technical University of Denmark
ansup@dtu.dk

**Kristoffer Stensbo-Smidt**
Technical University of Denmark
krss@dtu.dk

**Wouter Boomsma**
University of Copenhagen
wb@di.ku.dk

**Jes Frellsen**
Technical University of Denmark
jefr@dtu.dk

## Abstract

In variational inference, the benefits of Bayesian models rely on accurately capturing the true posterior distribution. We propose using neural samplers that specify implicit distributions, which are well-suited for approximating complex multimodal and correlated posteriors in high-dimensional spaces. Our approach advances inference using implicit distributions by introducing novel approximate bounds by locally linearising the neural sampler. This is distinct from existing methods that rely on additional discriminator networks and unstable adversarial objectives. Furthermore, we present a new sampler architecture that, for the first time, enables implicit distributions over tens of millions of latent variables, addressing computational concerns by using differentiable numerical approximations. Our empirical analysis indicates our method is capable of recovering correlations across layers in large Bayesian neural networks, a property that is crucial for a network's performance but notoriously challenging to achieve. To the best of our knowledge, no other method has been shown to accomplish this task for such large models. Through experiments on downstream tasks, we demonstrate that our expressive posteriors outperform state-of-the-art uncertainty quantification methods, validating the effectiveness of our training algorithm and the quality of the learned implicit distribution.

## 1 Introduction

In the Bayesian approach to statistical inference, information about the data is captured by the posterior distribution over a model's latent variables. By marginalising (averaging) over the variables weighted by the posterior, Bayesian inference can provide excellent generalisation and model calibration. This is particularly compelling for complex models, such as modern deep models, where extreme overparametrisation makes overfitting inevitable unless precaution is taken (Wilson et al., 2020).

Exact Bayesian inference is, however, intractable for all but the simplest models. Practitioners, therefore, rely on approximate inference frameworks, which can be broadly categorised as either sampling-based approaches or variational inference (VI, Saul et al., 1996; Peterson, 1987), the latter being the preferred approach for high-dimensional problems. VI seeks a tractable approximation to the intractable true posterior, limiting us to simple approximations like the isotropic Gaussian, which are often far too crude to satisfactorily cover the posterior, either only narrowly covering a single posterior mode or placing excessive mass in low-probability regions of the posterior (Foong et al., 2019, 2020).

37th Conference on Neural Information Processing Systems (NeurIPS 2023).

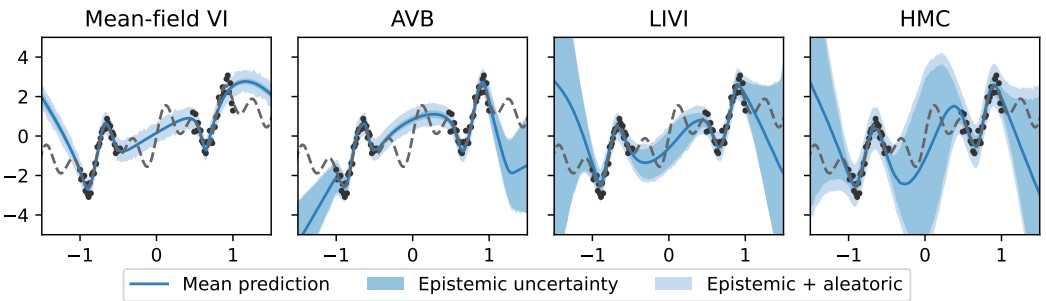

Figure 1: A Bayesian neural network trained on a small regression dataset using four different approximation methods: mean-field variational inference (VI), Adversarial Variational Bayes (AVB, Mescheder et al., 2017a), our proposed model (LIVI), and Hamiltonian Monte Carlo (HMC). The dashed line represents the true underlying function from which the training data (black points) were sampled. For each model, we show the mean prediction together with the epistemic uncertainty (dark shaded region) and the combined epistemic and aleatoric uncertainties (light shaded region).

To achieve more expressive approximations, *implicit variational inference* (IVI) defines distributions implicitly by transforming samples from a simple distribution. This provides much greater flexibility at the cost of not having a tractable likelihood for the posterior approximation (Diggle et al., 1984; Mohamed et al., 2016) and thus often requires unstable adversarial objectives based on density ratio estimation, which are not easy to scale to high dimensions (Sugiyama et al., 2012; Huszár, 2017).

The different approaches to posterior approximation can be imagined lying on a spectrum illustrated in Fig. 1. On the left, the mean-field VI approximation is much too confident in regions without data. Sampling-based approaches, here represented by Hamiltonian Monte Carlo (HMC) on the right, converge to the true posterior given enough compute, which, however, is typically prohibitively large. Implicit VI approaches, here represented by our proposed approximation, LIVI (Section 4), and Adversarial Variational Bayes (AVB, Mescheder et al., 2017a), which uses a discriminator network for density ratio estimation, hit a tractable middle-ground between the two extremes.

LIVI can produce highly flexible implicit densities approximating the posteriors of complex models through general (non-invertible) stochastic transformations. We present novel, stable training objectives through approximating a lower bound to the model evidence as well as a novel generator architecture, which results in a vastly improved capacity to model expressive distributions, even in millions of dimensions. Specifically, our contributions are:

- We derive a novel approximate lower bound for variational inference using an implicit variational approximation to avoid unstable adversarial objectives.

- To keep computational overhead at large scale in check, we further lower bound the approximate lower bound reducing memory requirements and run times significantly.

- We devise a novel generator architecture capable of specifying highly correlated distributions in high dimensions whilst using a very manageable number of weights and biases, a necessity for modelling high-dimensional distribution.

The paper is organised as follows: Section 2 introduces the background on VI for Bayesian models in general, and VI with implicit distributions (IVI) in particular. Section 3 introduces our proposals to tackle well-known challenges in IVI. Integrating these proposals in Section 4 yields our novel bounds for implicit VI, and Section 5 places our paper in the context of the published literature. In Section 6, we experimentally evaluate our proposed method before summarising and concluding in Section 7.

## 2 Variational inference for Bayesian models

Consider the supervised learning setting, where we have a training set $\mathcal{D} = \{(\boldsymbol{x}_i, \boldsymbol{y}_i)\}_{i=1}^n$, $\boldsymbol{X} = \{\boldsymbol{x}_i\}_{i=1}^n$ are the covariates (inputs) and $\boldsymbol{Y} = \{\boldsymbol{y}_i\}_{i=1}^n$ are the labels (outputs), which we wish to fit using a Bayesian model parametrised by $\boldsymbol{\theta} \in \mathbb{R}^m$. That is, assuming a prior $p(\boldsymbol{\theta})$ over the parameters and a likelihood $p(\mathcal{D} \,|\, \boldsymbol{\theta})$, we wish to compute the posterior $p(\boldsymbol{\theta} \,|\, \mathcal{D}) = \frac{p(\mathcal{D} \,|\, \boldsymbol{\theta})p(\boldsymbol{\theta})}{p(\mathcal{D})}$. As

computing this posterior is intractable for all but the simplest models, we turn to the framework of *variational inference* (VI) which finds the best matching tractable *variational* distribution, $q_{\boldsymbol{\gamma}}(\boldsymbol{\theta})$ parametrised by $\boldsymbol{\gamma}$ to the true posterior, $p(\boldsymbol{\theta} \,|\, \mathcal{D})$. A sensible measure of their closeness is the (intractable) Kullback-Leibler (KL) divergence between the two distributions defined as

$$\mathrm{D}_{\mathrm{KL}}\big[q_{\boldsymbol{\gamma}}(\boldsymbol{\theta}) \,\big\|\, p(\boldsymbol{\theta} \,|\, \mathcal{D})\big] = \log p(\mathcal{D}) - \underbrace{\mathbb{E}_{q_{\boldsymbol{\gamma}}(\boldsymbol{\theta})}\big[\log p(\boldsymbol{\theta}, \mathcal{D}) - \log q_{\boldsymbol{\gamma}}(\boldsymbol{\theta})\big]}_{\mathcal{L}(\boldsymbol{\gamma})}. \tag{1}$$

Since the evidence, $p(\mathcal{D})$, does not depend on $\boldsymbol{\gamma}$, minimising the above divergence is equivalent to maximising $\mathcal{L}(\boldsymbol{\gamma})$, referred to as the evidence lower bound (ELBO), which then becomes the objective function in VI. It can be further expanded into a *likelihood term* and a *regularisation term* as

$$\mathcal{L}(\boldsymbol{\gamma}) = \underbrace{\mathbb{E}_{q_{\boldsymbol{\gamma}}(\boldsymbol{\theta})}\big[\log p(\mathcal{D} \,|\, \boldsymbol{\theta})\big]}_{\text{likelihood term}} - \underbrace{\mathrm{D}_{\mathrm{KL}}\big[q_{\boldsymbol{\gamma}}(\boldsymbol{\theta}) \,\big\|\, p(\boldsymbol{\theta})\big]}_{\text{regularisation term}}, \tag{2}$$

where the likelihood term encourages the variational approximation to model the data well, and the regularisation term tries to keep the posterior close to the prior.

## 2.1 Implicit variational inference

In implicit VI (IVI), the variational distribution is only implicitly defined through its generative process which allows sampling from this distribution, but neither its density nor the gradients of its density can be evaluated. One way to draw from an implicit distribution is to push a low-dimensional noise vector through a neural network (Mohamed et al., 2016; Tran et al., 2017; Shi et al., 2018). These models have also been called amortised neural samplers (Kumar et al., 2019), with the generative process $\boldsymbol{z} \sim q(\boldsymbol{z})$ and $\boldsymbol{\theta} = g_{\boldsymbol{\gamma}}(\boldsymbol{z})$ and the corresponding density

$$q_{\boldsymbol{\gamma}}(\boldsymbol{\theta}) = \frac{\partial}{\partial \theta_1} \frac{\partial}{\partial \theta_2} \cdots \frac{\partial}{\partial \theta_m} \int_{\boldsymbol{z} \,|\, g_{\boldsymbol{\gamma}}(\boldsymbol{z}) \leq \boldsymbol{\theta}} q(\boldsymbol{z}) \, \mathrm{d}\boldsymbol{z}, \tag{3}$$

where $q(\boldsymbol{z})$ is a fixed base distribution and $g_{\boldsymbol{\gamma}} : \mathbb{R}^d \to \mathbb{R}^m$ is a non-linear, typically non-invertible mapping due to which this integral is non-trivial to solve. The resulting intractability of the density precludes the usage of popular objectives in probabilistic modelling. The *likelihood term* from Eq. (2) and its gradients can be estimated using Monte Carlo and the reparameterization trick (Kingma et al., 2014). However, the *regularisation term* involves the entropy of $q_{\boldsymbol{\gamma}}$, $H_q[q_{\boldsymbol{\gamma}}]$:

$$\mathrm{D}_{\mathrm{KL}}\big[q_{\boldsymbol{\gamma}}(\boldsymbol{\theta}) \,\big\|\, p(\boldsymbol{\theta})\big] = \mathbb{E}_{\boldsymbol{\theta} \sim q_{\boldsymbol{\gamma}}(\boldsymbol{\theta})}\left[\log \frac{q_{\boldsymbol{\gamma}}(\boldsymbol{\theta})}{p(\boldsymbol{\theta})}\right] = \underbrace{\mathbb{E}_{\boldsymbol{\theta} \sim q_{\boldsymbol{\gamma}}(\boldsymbol{\theta})}\big[\log q_{\boldsymbol{\gamma}}(\boldsymbol{\theta})\big]}_{-H_q[q_{\boldsymbol{\gamma}}]} - \mathbb{E}_{\boldsymbol{\theta} \sim q_{\boldsymbol{\gamma}}(\boldsymbol{\theta})}\big[\log p(\boldsymbol{\theta})\big], \tag{4}$$

which following the intractability of the density Eq. (3) is not explicitly available. Moreover, critical to consider, $q_{\boldsymbol{\gamma}}$ is not a well-defined density in the parameter space, as its support lies on a low dimensional manifold and has measure zero. Consequently, the KL divergence in Eq. (1) is not always well-defined (Arjovsky et al., 2017) giving rise to another noteworthy issue with IVI. To tackle this issue, standard IVI approaches (Sugiyama et al., 2012; Huszár, 2017) rely on density ratio estimators based on a discriminator (in GANs) to estimate the *regularisation term*. Geng et al. (2021) give a tractable and differentiable lower bound on this entropy.

## 3 A deep latent variable model and its entropy

In this section, we address the issues noted above, **I**: The implicit density lies on a low dimensional manifold, leading to ill-defined KL (Eq. (1)) and **II**: The entropy of the implicit density and its gradients are intractable.

## 3.1 Defining a proper KL divergence

To make the KL in Eq. (1) well-defined, we add Gaussian noise to the output of our neural sampler $g_{\boldsymbol{\gamma}}$ which has the effect of making the implicit density continuous in the ambient space of $\boldsymbol{\theta}$, see Arjovsky et al. (Lemma 1, 2017) . Accordingly, we switch to the corresponding variational distribution – a Gaussian deep latent variable model (DLVM). This is a special case of the semi-implicit distribution

(Yin et al., 2018; Titsias et al., 2019) and equivalent to the generative model in a Gaussian variational autoencoder (VAE, Kingma et al., 2014; Rezende et al., 2014):

$$q_{\boldsymbol{\gamma}}(\boldsymbol{\theta}) = \int q_{\boldsymbol{\gamma}}(\boldsymbol{\theta} \,|\, \boldsymbol{z}) q(\boldsymbol{z}) \, \mathrm{d}\boldsymbol{z} = \mathbb{E}_{\boldsymbol{z} \sim q(\boldsymbol{z})}[q_{\boldsymbol{\gamma}}(\boldsymbol{\theta} \,|\, \boldsymbol{z})], \quad \text{where,} \tag{5}$$

$$q_{\boldsymbol{\gamma}}(\boldsymbol{\theta} \,|\, \boldsymbol{z}) = \mathcal{N}(\boldsymbol{\theta} \,|\, g_{\boldsymbol{\gamma}}(\boldsymbol{z}), \sigma^2 \boldsymbol{I}_m), \quad g_{\boldsymbol{\gamma}} : \mathbb{R}^d \to \mathbb{R}^m, \tag{6}$$

$\boldsymbol{\theta} \in \mathbb{R}^m$ and the latent variable $\boldsymbol{z} \in \mathbb{R}^d$. We assume a Gaussian base density $q(\boldsymbol{z}) = \mathcal{N}(\boldsymbol{z} \,|\, \boldsymbol{0}, \boldsymbol{I}_d)$. $g_{\boldsymbol{\gamma}}$ is the neural sampler (or generator) and $\sigma^2 \in \mathbb{R}^+$ is the fixed, homoscedastic variance of the output density, which we take to be small. Thus, each sample from the base density, $\boldsymbol{z}$, results in a Gaussian output density, $q_{\boldsymbol{\gamma}}(\boldsymbol{\theta} \,|\, \boldsymbol{z})$. And the marginalisation leads to a very flexible $q_{\boldsymbol{\gamma}}(\boldsymbol{\theta})$ which informally, is an infinite mixture of Gaussians (example 2, Im et al., 2017). Generally, we do not have a closed form for $q_{\boldsymbol{\gamma}}(\boldsymbol{\theta})$ and consequently $H[q_{\boldsymbol{\gamma}}]$, due to the non-linear function $g_{\boldsymbol{\gamma}}$ within $q_{\boldsymbol{\gamma}}(\boldsymbol{\theta} \,|\, \boldsymbol{z})$ in Eq. (5), so we are still left with challenge **II**, which we address next.

## 3.2 Approximating the intractable entropy

With the DLVM, we can expand the entropy of the variational approximation in Eq. (4) as,

$$H[q_{\boldsymbol{\gamma}}(\boldsymbol{\theta})] = -\mathbb{E}_{\boldsymbol{z} \sim q(\boldsymbol{z})} \mathbb{E}_{\boldsymbol{\theta} \sim q_{\boldsymbol{\gamma}}(\boldsymbol{\theta} \,|\, \boldsymbol{z})}[\log q_{\boldsymbol{\gamma}}(\boldsymbol{\theta})]. \tag{7}$$

In principle, a Monte Carlo estimator of Eq. (5) could be used to approximate the $q_{\boldsymbol{\gamma}}(\boldsymbol{\theta})$ term and by extension, this entropy, but this estimator is biased and has high variance. This variance can be reduced with directed sampling via an encoder, but to avoid training an additional network, we derive an encoder-free approximation in the following. To do this, we need to work around the non-linearity of $g_{\boldsymbol{\gamma}}$ that causes the intractability.

**Linearisation of the generator** First, to estimate $q_{\boldsymbol{\gamma}}(\boldsymbol{\theta})$ during training with lower variance than a direct Monte Carlo estimate, we consider an analytical approximation of Eq. (7) obtained via a local linearisation of the generator/neural sampler around $\boldsymbol{z}'$:

$$T^1_{\boldsymbol{z}'}(\boldsymbol{z}) = g_{\boldsymbol{\gamma}}(\boldsymbol{z}') + \boldsymbol{J}_g(\boldsymbol{z}') \, (\boldsymbol{z} - \boldsymbol{z}'), \tag{8}$$

where $\boldsymbol{J}_g(\boldsymbol{z}') \in \mathbb{R}^{m \times d}$ is the Jacobian of $g_{\boldsymbol{\gamma}}$ evaluated in $\boldsymbol{z}'$, which we assume exists. We can approximate $g_{\boldsymbol{\gamma}}(\boldsymbol{z})$ by the linear function $T^1_{\boldsymbol{z}'}(\boldsymbol{z})$ when $\boldsymbol{z}$ is close to $\boldsymbol{z}'$ and obtain a Gaussian approximation of the output density, $q_{\boldsymbol{\gamma}}(\boldsymbol{\theta} \,|\, \boldsymbol{z}) \approx \tilde{q}_{\boldsymbol{z}'}(\boldsymbol{\theta} \,|\, \boldsymbol{z}) = \mathcal{N}(\boldsymbol{\theta} \,|\, T^1_{\boldsymbol{z}'}(\boldsymbol{z}), \sigma^2 \boldsymbol{I}_m)$ and substitute this into Eq. (5) to approximate $q_{\boldsymbol{\gamma}}(\boldsymbol{\theta})$:

$$q_{\boldsymbol{\gamma}}(\boldsymbol{\theta}) = \mathbb{E}_{\boldsymbol{z} \sim q(\boldsymbol{z})}[q_{\boldsymbol{\gamma}}(\boldsymbol{\theta} \,|\, \boldsymbol{z})] \approx \mathbb{E}_{\boldsymbol{z} \sim q(\boldsymbol{z})}[\tilde{q}_{\boldsymbol{z}'}(\boldsymbol{\theta} \,|\, \boldsymbol{z})] \tag{9}$$

$$= \mathbb{E}_{\boldsymbol{z} \sim q(\boldsymbol{z})}[\mathcal{N}(\boldsymbol{\theta} \,|\, g_{\boldsymbol{\gamma}}(\boldsymbol{z}') + \boldsymbol{J}_g(\boldsymbol{z}') \, (\boldsymbol{z} - \boldsymbol{z}'), \sigma^2 \boldsymbol{I}_m)] \tag{10}$$

Now the expectation in Eq. (10) can be analytically solved by integrating over the latent variable (Tipping et al., 1999) and is Gaussian, which gives the closed form of the approximation

$$q_{\boldsymbol{\gamma}}(\boldsymbol{\theta}) \approx \mathcal{N}(\boldsymbol{\theta} \,|\, \boldsymbol{\mu}_{\boldsymbol{\gamma}}(\boldsymbol{z}'), C_{\boldsymbol{\gamma}}(\boldsymbol{z}')) =: \tilde{q}_{\boldsymbol{z}'}(\boldsymbol{\theta}), \tag{11}$$

where

$$\boldsymbol{\mu}_{\boldsymbol{\gamma}}(\boldsymbol{z}') = g_{\boldsymbol{\gamma}}(\boldsymbol{z}') - \boldsymbol{J}_g(\boldsymbol{z}') \, \boldsymbol{z}', \qquad C(\boldsymbol{z}') = \boldsymbol{J}_g(\boldsymbol{z}') \boldsymbol{J}_g(\boldsymbol{z}')^\mathsf{T} + \sigma^2 \boldsymbol{I}_m. \tag{12}$$

**Approximation of the differential entropy** We use the above result to approximate the entropy of the DLVM,

$$H[q_{\boldsymbol{\gamma}}(\boldsymbol{\theta})] = -\mathbb{E}_{\boldsymbol{z} \sim q(\boldsymbol{z})} \mathbb{E}_{\boldsymbol{\theta} \sim q_{\boldsymbol{\gamma}}(\boldsymbol{\theta} \,|\, \boldsymbol{z})}[\log q_{\boldsymbol{\gamma}}(\boldsymbol{\theta})] \approx -\mathbb{E}_{\boldsymbol{z} \sim q(\boldsymbol{z})} \mathbb{E}_{\boldsymbol{\theta} \sim q_{\boldsymbol{\gamma}}(\boldsymbol{\theta} \,|\, \boldsymbol{z})}[\log \tilde{q}_{\boldsymbol{z}'=\boldsymbol{z}}(\boldsymbol{\theta})], \tag{13}$$

where $\log \tilde{q}_{\boldsymbol{z}'=\boldsymbol{z}}(\boldsymbol{\theta})$ is Eq. (11) evaluated at the samples $\boldsymbol{z}$ from the outer expectation. Importantly, we do the linearisation of $q_{\boldsymbol{\gamma}}(\boldsymbol{\theta})$ around the latent value $\boldsymbol{z}$ that is used to sample each $\boldsymbol{\theta}$ in the expectation, which means that for small $\sigma^2$-values, $\boldsymbol{z}$ is close to $\boldsymbol{z}'$, and the linear approximation will be good. This amounts to using a point-wise Gaussian to approximate the entropy at $\boldsymbol{z}$ in the training objective, whilst sampling from the DLVM noted in Eq. (5). Hence, important to consider here, that the linearisation does not restrict the expressivity of the posterior that we finally obtain.

We could, in principle, evaluate the approximation in Eq. (13) numerically, but it involves the matrix inverse of $C(\boldsymbol{z}')$, which is computationally demanding. In Appendix A, we show that for small values of the output variance $\sigma^2$, we can further approximate the differential entropy by

$$H[q_{\boldsymbol{\gamma}}(\boldsymbol{\theta})] \approx \frac{1}{2} \mathbb{E}_{\boldsymbol{z} \sim q(\boldsymbol{z})} \left[ \log \det \left( \boldsymbol{J}_g(\boldsymbol{z}) \boldsymbol{J}_g(\boldsymbol{z})^\mathsf{T} + \sigma^2 \boldsymbol{I}_m \right) \right] + \frac{m}{2} + \frac{m}{2} \log 2\pi \tag{14}$$

# 4 Linearised Implicit Variational Inference

Finally, we arrive at our novel approximation to the ELBO for IVI using Eq. (5) as the variational distribution. Using the entropy approximation from Eq. (14), we obtain the approximate ELBO denoted the *Linearised Implicit Variational Inference (LIVI) bound*

$$\mathcal{L}'(\gamma) = \mathbb{E}_{\boldsymbol{\theta} \sim q_{\boldsymbol{\gamma}}(\boldsymbol{\theta})} \left[ \log p(\mathcal{D} \,|\, \boldsymbol{\theta}) + \log p(\boldsymbol{\theta}) \right] + \frac{1}{2} \mathbb{E}_{\boldsymbol{z} \sim q(\boldsymbol{z})} \left[ \log \det \left( \boldsymbol{J}_g(\boldsymbol{z}) \boldsymbol{J}_g(\boldsymbol{z})^\intercal + \sigma^2 \boldsymbol{I}_m \right) \right] + c, \tag{15}$$

where $c = \frac{m}{2} + \frac{m}{2} \log 2\pi$. As the determinant of a matrix is the product of its singular values, the log determinant is the sum of the log of these singular values. If $s_d(\boldsymbol{z}) \geq \ldots \geq s_1(\boldsymbol{z})$ are the non-zero singular values of the Jacobian $\boldsymbol{J}_g(\boldsymbol{z})$, we can write the log determinant term as

$$\frac{1}{2} \log \det(\boldsymbol{J}_g(\boldsymbol{z}) \boldsymbol{J}_g(\boldsymbol{z})^\intercal + \sigma^2 \boldsymbol{I}_m) = \frac{1}{2} \sum_{i=1}^{d} \log(s_i^2(\boldsymbol{z}) + \sigma^2) + \frac{m-d}{2} \log \sigma^2 \tag{16}$$

To avoid calculating all the singular values of a large Jacobian matrix, we can follow Geng et al. (2021, Eq. 10) and lower-bound Eq. (16) as

$$\frac{1}{2} \sum_{i=1}^{d} \log(s_i^2(\boldsymbol{z}) + \sigma^2) + \frac{m-d}{2} \log \sigma^2 \geq \frac{d}{2} \log(s_1^2(\boldsymbol{z}) + \sigma^2) + \frac{m-d}{2} \log \sigma^2 \tag{17}$$

We defer the details of Eq. (16) and the algorithm (LOBPCG, Knyazev, 2001) for obtaining $s_1$ to Appendix C. This allows us to extend our variational approximation to high-dimensional parameter spaces, and using Eq. (17) we obtain a lower bound on $\mathcal{L}'(\gamma)$ given by

$$\mathcal{L}''(\gamma) = \mathbb{E}_{\boldsymbol{\theta} \sim q_{\boldsymbol{\gamma}}(\boldsymbol{\theta})} \left[ \log p(\mathcal{D} \,|\, \boldsymbol{\theta}) + \log p(\boldsymbol{\theta}) \right] + \mathbb{E}_{\boldsymbol{z} \sim q(\boldsymbol{z})} \left[ \frac{d}{2} \log(s_1^2(\boldsymbol{z}) + \sigma^2) \right] + c. \tag{18}$$

We denote $\mathcal{L}'(\gamma)$ the approximate LIVI bound with whole Jacobians and $\mathcal{L}''(\gamma)$ the LIVI bound with a differentiable lower bound on the determinant. We draw comparisons between how well these bounds estimate the entropy of a DLVM model in Appendix F.5, and write the reparameterised version of both bounds in Appendix B. The relationships between the bounds and the evidence are

$$\log p(\mathcal{D}) \geq \mathcal{L}(\gamma) \approx \mathcal{L}'(\gamma) \geq \mathcal{L}''(\gamma) \tag{19}$$

Based on the compute budget, the two LIVI bounds offer an accuracy vs compute trade-off. In both cases, entropy maximisation promotes the exploration of the latent space. It is also worthwhile to note that Eq. (15), which uses the entropy approximation in Eq. (13), is fundamentally different from other linearisation approaches in the literature (e.g., Immer et al., 2021), as we do not linearise the Bayesian model. Instead, our approach linearises the generator parameterising $q_{\boldsymbol{\gamma}}(\boldsymbol{\theta} \,|\, \boldsymbol{z})$ only when needed to approximate the differential entropy, thus leaving the variational approximation unchanged.

# 5 Related Work

The usage of a secondary model to generate parameters of a primary model first appeared in the form of *hypernetworks* (Ha et al., 2017). Our approach is probabilistic and is hence closer to Bayesian hypernetworks (Krueger et al., 2017) that mandate invertibility of the generator. The formulation is exactly like normalising flows and thereby sidesteps the complexities of estimating the entropy term. Flow-based approximations require particular focus on the design of the variational approximation to curb the dimensionality of the flow and the Jacobian matrices. Louizos et al. (2017) use an expressive flow on the multiplicative factors of the weights in each layer and not on all weights jointly. Our method does not necessitate invertibility, making it more general. We also share parallels with works in *manifold learning* (Brehmer et al., 2020; Caterini et al., 2021) in that they also use noninvertible mappings and have to employ a general form of the standard change of variable formula $\log \det(\boldsymbol{J}_g(\boldsymbol{z})^\intercal \boldsymbol{J}_g(\boldsymbol{z}))$ in the approximating density. Kristiadi et al. (2022) infers an expressive distribution over a subset of weights by using a *post hoc* Laplace approximation as the base distribution for a normalising flow. This approach, however, inherits the usual scalability limitations of flows.

Broadly, works in implicit VI have proposed novel ways of estimating the ratio of the variational approximation to the prior (regularisation-term), usually referred to as density-ratio estimation (also referred to as the prior-contrastive formulation by Huszár, 2017). To the best of our knowledge, we are the first to propose and test an entropy approximation for an implicit density used as a variational approximation. In particular, Shi et al. (2018), Tran et al. (2017), and Pawlowski et al. (2017) have successfully demonstrated implicit variational inference using hypernetworks and align well with our goals. Tran et al. (2017) opt for training a discriminator network to maximally distinguish two distributions given only i.i.d. samples from each. This approach, though general, adds to the computational requirements and becomes more challenging in high dimensions (Sugiyama et al., 2012). To mitigate the overhead of training the discriminator for each update of the ELBO, many works limit the discriminator training to a single or few iterations. Furthermore, this approach entails an adversarial objective that is notoriously unstable (Mescheder et al., 2017b). Pawlowski et al. (2017) treat all weights as independent and find that a single discriminator network is inaccurate at estimating log ratios compared to the analytical form of *Bayes by backprop* (Blundell et al., 2015), and opt to use a kernel method that matches the analytical form more closely.

Shi et al. (2018) propose a novel way of estimating the ratio of the two densities using kernel regression in the space of parameters which obviates the need for a min-max objective but we expect to be inaccurate in estimating high-dimensional density ratios especially given a limited number of samples from both the densities as well as the RBF kernel. Pradier et al. (2018) are also motivated by the possibility of compressing the posterior in a lower-dimensional space and consider the parameters of the generator to be stochastic. They use an inference network with the generator/decoder and hence require empirical latent samples to train which doubles the training steps and limits the scalability. SIVI (Yin et al., 2018) and D-SIVI (Molchanov et al., 2019) use Monte Carlo (MC) averaging to approximate the variational approximation hence precluding an approximation over an intractable entropy as their resultant ELBOs only contains the explicit conditional $q_\gamma(\theta \mid z)$. Their novelties lie in using a general semi-implicit formulation to model all the weights of the network but as noted earlier this MC estimator is biased and has high variance. Yin et al., 2018 propose upper and lower bounds on the ELBO to train with this variational approximation and Molchanov et al. (2019) extend the approach by using an implicit prior.

Recently, several works have considered alternatives to high-dimensional inference by heuristically limiting randomness, e.g., Izmailov et al. (2020), Daxberger et al. (2021a,b), and Sharma et al. (2023). Our work is somewhat orthogonal to these, as we present a general method for performing high-dimensional inference. Indeed, while our focus here has been on inference in full networks, our bounds can readily be applied to subspaces or subnetworks too. A notable difference from the works building on the Laplace approximation, however, is that these works linearise the model during prediction and inference, whereas we do not. We linearise the neural sampler, $g_\gamma(z)$, but only when it is used to estimate the intractable entropy and its gradients, see Eqs. (10) to (12). Thus, the resulting posterior approximation is, still a highly flexible (i.e., non-linearised) implicit distribution.

## 6 Experiments

We now test the proposed posterior approximation (Section 4) for its generalisation and calibration capabilities in downstream tasks on different established datasets. We use Bayesian Neural Networks (BNNs) for our analysis due to the immense number of global latent variables (i.e., the weights and biases) that are required by modern BNN architectures to perform well with larger datasets, validating our method on previously unreachable and untested scales for implicit VI.

We evaluate multiple posterior approximation schemes for solving various downstream tasks, with the final performance of the BNN serving as a metric for assessing the quality of each method's approximation of the true posterior. Ideally, we wish to understand whether an expressive posterior approximation helps uncertainty quantification in downstream tasks. As this is a challenging question to address directly, we compare LIVI against less flexible baselines to determine whether capturing posterior complexity leads to improved predictive performance and uncertainty estimates.

### 6.1 Experimental setup

**Model definition**   Briefly, BNNs are standard neural networks with a prior on their parameters, which, post-training, results in a posterior distribution. In a notation similar to that of Section 2, BNNs can be presented as $p(\theta \mid (\boldsymbol{x}, y)) \propto p(y \mid f_\theta(\boldsymbol{x}))p(\theta)$, where $\theta$ represents the parameters of a

neural network denoted by $f$. The likelihood, $p(y \mid f_{\boldsymbol{\theta}})$, depends on the given task. In this section, we test proxies to the true posterior $p(\boldsymbol{\theta} \mid (\boldsymbol{x}, y))$ for how well they 1) model the dataset, and 2) model regions away from the dataset.

For regression tasks, we use a Gaussian likelihood and learn the homoscedastic variance, $\eta^2$, using type-II maximum likelihood, $p(y \mid f_{\boldsymbol{\theta}}(\boldsymbol{x})) = \mathcal{N}(y \mid \mu = f_{\boldsymbol{\theta}}(\boldsymbol{x}), \eta^2)$. For classification tasks, we use a categorical likelihood function for the labels with logits produced by $f_{\boldsymbol{\theta}}$.

**Generator architecture**    To efficiently generate millions of correlated parameters, we introduce a novel architecture for the generator $g_{\boldsymbol{\gamma}}(\boldsymbol{z})$. We sample a low-dimensional noise matrix, which is gradually scaled up through the generator network using matrix multiplications (Shi et al., 2018). To generate all the parameters of the BNN, we gradually scale up this noise through the generator. The output from the first matrix multiplication layer is split and fed into a series of disconnected matrix multiplication networks that produce parameters for the individual layers of the BNN. This equates to pruning dense connections deeper in the network. This architecture is designed such that it can capture both within-layer correlations in the BNN (through the individual subnetworks), and across-layer correlations (through the first matrix multiplication), whilst having a manageable number of parameters. The architecture is described in detail in Appendix D and has been used for experiments in Sections 6.5 and 6.6.

**Method comparison**    For LIVI, the generator or hypernetwork, $g_{\boldsymbol{\gamma}}(\boldsymbol{z})$, represents the implicit distribution over network parameters for the BNN, which we sample from. We explicitly mention which of the ELBOs, Eqs. (15) and (16), we use in every experiment. We compare our method with other scalable uncertainty quantification methods: Adversarial Variational Bayes (AVB, Mescheder et al., 2017a), last-layer Laplace approximation (LLLA, Daxberger et al., 2021a), deep ensembles (DE, Lakshminarayanan et al., 2017), mean-field VI (MFVI, Saul et al., 1996; Osawa et al., 2019), as well as a simple MAP estimate. We chose DEs over other methods, such as SWAG (Maddox et al., 2019), as Daxberger et al. (2021a) found this to be a stronger baseline. Where possible, we compare against Kernel Implicit Variational Inference (KIVI, Shi et al., 2018); these results are taken from their paper. The only other implicit VI method is AVB, which, in all experiments, uses the exact same generator architecture as LIVI. Where feasible, we compare with HMC as it is the gold-standard posterior approximation technique.

**Overview of experiments**    The first two sections focus on regression; a toy example in Section 6.2 and tests on UCI regression datasets in Section 6.3. Next, we move to classification experiments, focusing on out-of-distribution (OOD) benchmarks. Section 6.4 shows results on MNIST, then Section 6.5 considers images from CIFAR10, allowing us to perform OOD testing of the posterior approximations in millions of dimensions. In Section 6.6, we test on CIFAR100, evaluating our approach on even larger dimensional BNN posteriors [1]. We discuss the metrics used in Appendix E.1.

## 6.2   Toy data

In Fig. 1, we compare posterior predictives of our method (LIVI) against the standard baselines for posterior inference on a simple sinusoidal toy dataset. We qualitatively assess the epistemic uncertainty, or the variance in the function space, in regions away from training data. The BNN contains 105 parameters and is thus small enough that we can use $\mathcal{L}'$, Eq. (15), as the objective function for LIVI, (for the remaining details of the architectures and the training, see Appendix E.3). We observe that both AVB and MFVI dramatically underestimate uncertainties, whereas LIVI is much closer to Hamiltonian Monte Carlo (HMC). We further observe that LIVI's learned posterior approximation has multiple modes and heavy tails, see Appendix F.4.

## 6.3   UCI datasets

Here, we follow the setups in Lakshminarayanan et al. (2017) and Shi et al. (2018) and choose a single hidden-layered MLP as the BNN for all datasets. Due to the different dimensionalities of the datasets, the number of BNN parameters vary between 501 and 751. Still, at this scale, it is feasible to work with whole Jacobian matrices, which allows us to test $\mathcal{L}'$, Eq. (15), as the objective function and

---

[1]Supporting code is available here - `https://github.com/UppalAnshuk/LIVI_neurips23`

Table 1: **UCI regression datasets.** We report RMSE (↓) and log-likelihoods (↑) on the test set and average across three different seeds for each model to quantify the variance in the results.

| | Method | Boston | Concrete | Energy | Kin8nm | Naval |
|---|---|---|---|---|---|---|
| **Test RMSE** | LIVI ($\mathcal{L}'$) | $2.32 \pm 0.07$ | $\mathbf{4.24 \pm 0.17}$ | $0.41 \pm 0.27$ | $\mathbf{0.03 \pm 0.00}$ | $\mathbf{0.00 \pm 0.00}$ |
| | LIVI ($\mathcal{L}''$) | $2.40 \pm 0.09$ | $4.62 \pm 0.13$ | $0.44 \pm 0.11$ | $0.08 \pm 0.01$ | $0.00 \pm 0.01$ |
| | HMC | $\mathbf{2.26 \pm 0.00}$ | $4.27 \pm 0.00$ | $\mathbf{0.38 \pm 0.00}$ | $0.04 \pm 0.00$ | $\mathbf{0.00 \pm 0.00}$ |
| | DE | $3.28 \pm 1.00$ | $6.03 \pm 0.58$ | $2.09 \pm 0.29$ | $0.09 \pm 0.00$ | $0.00 \pm 0.00$ |
| | KIVI | $2.80 \pm 0.17$ | $4.70 \pm 0.12$ | $0.47 \pm 0.02$ | $0.08 \pm 0.00$ | $0.00 \pm 0.00$ |
| **Test LL** | LIVI ($\mathcal{L}'$) | $\mathbf{-2.16 \pm 0.05}$ | $-2.79 \pm 0.11$ | $-1.17 \pm 0.13$ | $1.24 \pm 0.04$ | $6.74 \pm 0.04$ |
| | LIVI ($\mathcal{L}''$) | $-2.40 \pm 0.09$ | $-2.99 \pm 0.13$ | $-1.37 \pm 0.11$ | $1.15 \pm 0.01$ | $5.84 \pm 0.06$ |
| | HMC | $-2.20 \pm 0.00$ | $\mathbf{-2.67 \pm 0.00}$ | $\mathbf{-1.14 \pm 0.00}$ | $1.27 \pm 0.00$ | $\mathbf{7.79 \pm 0.00}$ |
| | DE | $-2.41 \pm 0.25$ | $-3.06 \pm 0.18$ | $-1.31 \pm 0.22$ | $\mathbf{1.28 \pm 0.02}$ | $5.93 \pm 0.05$ |
| | KIVI | $-2.53 \pm 0.10$ | $-3.05 \pm 0.04$ | $-1.30 \pm 0.01$ | $1.16 \pm 0.01$ | $5.50 \pm 0.12$ |

Table 2: **OOD Test M-C1:** We report out-of-distribution performance for models trained on MNIST (and evaluated on FMNIST, KMNIST and EMNIST) or CIFAR10 (and evaluated on SVHN, LSUN and CIFAR100). The metrics have been averaged across the OOD datasets.

| | Confidence ↓ | | AUROC ↑ | |
|---|---|---|---|---|
| Method | MNIST | CIFAR10 | MNIST | CIFAR10 |
| MAP | $72.10 \pm 0.36$ | $81.40 \pm 0.16$ | $96.32 \pm 0.22$ | $86.73 \pm 0.64$ |
| MFVI | $69.23 \pm 0.24$ | $74.71 \pm 0.23$ | $96.53 \pm 0.16$ | $87.50 \pm 0.25$ |
| LLLA | $67.40 \pm 0.19$ | $53.6 \pm 0.3$ | $96.67 \pm 0.27$ | $89.03 \pm 0.51$ |
| DE | $63.14 \pm 0.11$ | $67.17 \pm 0.21$ | $97.52 \pm 0.08$ | $89.61 \pm 0.11$ |
| LIVI | $\mathbf{55.03 \pm 0.13}$ | $\mathbf{43.47 \pm 0.28}$ | $\mathbf{97.91 \pm 0.27}$ | $\mathbf{91.83 \pm 0.41}$ |
| AVB | $70.68 \pm 0.45$ | NA | $95.5 \pm 0.4$ | NA |

to analyse the performance difference with $\mathcal{L}''$, Eq. (18). Further details of the generator architectures and baseline methods are presented in Appendix E.4.

Our results are summarised in Table 1. Both LIVI bounds, $\mathcal{L}'$ and $\mathcal{L}''$, perform similarly, giving empirical justification for the lower bound in Eq. (17). Furthermore, both bounds are comparable to HMC. We train our objective with far fewer samples compared to KIVI and outperform it for even small hypernetwork architectures.

## 6.4   MNIST Dataset

We use three different out-of-distribution tests to examine the estimated uncertainty in downstream tasks. We use the LeNet architecture for the Bayesian neural network with a total of $44\,000$ parameters, necessitating the use of our second objective $\mathcal{L}''$, Eq. (18). We train on MNIST, achieving an in-distribution test accuracy of $\sim 99.1\%$ for all methods except MFVI, which achieves $98.6\%$.

**OOD Test M-C1**   In this experiment, we consider three OOD datasets, FMNIST, KMNIST, EMNIST and Table 2 reports the averaged results. We outperform all the methods we compare against in both categories. This signifies that the implicit approximation makes the BNN less over-confident than competing methods when predicting over datasets that the BNN has not encountered in training.

**OOD Test M2**   We compare our method on another OOD benchmark by rotating the MNIST image by different angles (Daxberger et al., 2021a). For this benchmark, we plot the negative log-likelihoods and expected calibration errors for all rotation angles in Fig. 2, testing all approaches at varying extents of drift. We conclude that we perform the best in these two categories as well.

**OOD Test M3**   As the last benchmark on MNIST, we opt to plot the empirical CDF of predictive entropies across OOD images (Lakshminarayanan et al., 2017; Louizos et al., 2017). Under covariate shift, a well-calibrated model should predict a uniform distribution for never-before-seen data. So, with a model trained on MNIST we calculate predictive entropies on OOD datasets, plotting each method's fraction of high-entropy predictions in Fig. 3. In other words, the smaller the area between the curve and the dashed line, the better the method. We find that LIVI consistently produces

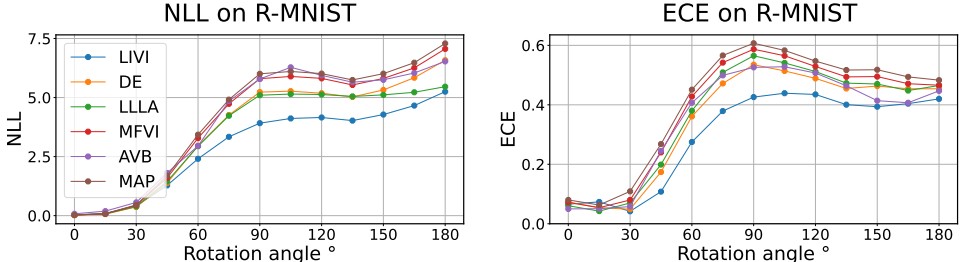

Figure 2: **OOD Test M2: Rotated MNIST benchmark.** LIVI performs as well as, or better, than competing methods.

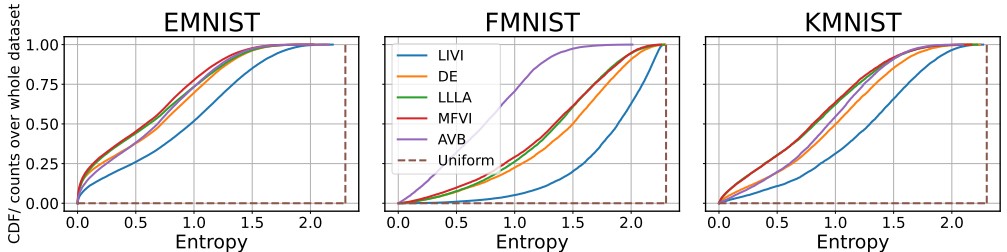

Figure 3: **OOD Test M3: Empirical CDF plot.** On the $x$-axis is the entropy of a single prediction and empirical CDF on the $y$-axis. With this plot, we count the number of high entropy (uniform distribution) predictions that each method makes on OOD data. As we count over the whole dataset, we would like to encounter more high entropy predictions closer to the value 2.30 on the $x$-axis.

higher-entropy predictions than competitors, showing that it is less overconfident for any OOD image.

We elaborate on our experimental set-ups in Appendix E.5, keeping the critical training parameters the same across methods and also reporting our computational requirements for these experiments.

## 6.5 CIFAR10

To test our approach at a larger scale, we consider the CIFAR10 image classification dataset. Here, for the BNN we use the same WideResNet architecture as Daxberger et al. (2021a) with 16 layers and a widen factor of 4 (Zagoruyko et al., 2016). This network architecture contains over 2.7 million trainable parameters, meaning that we again use our second objective $\mathcal{L}''$, Eq. (18). We generate all parameters with our novel hypernetwork architecture, described in Appendix D, with training details in Appendix E.6. All the methods reached very similar accuracies of $\sim 91.5\%$ on the in-distribution test set when trained for 200 epochs, except for MFVI, which reached $\sim 88.2\%$.

**OOD Test M-C1** We report averaged AUROC and confidence to test if the models assign wrong labels with high probability, see Table 2. The set of OOD datasets contains SVHN, LSUN, and CIFAR100. LIVI again outperforms competitors by being less overconfident about OOD predictions.

**OOD Test C2** We test the models' performances when presented with corrupted images from CIFAR-10-C (Hendrycks et al., 2019). The negative log-likelihoods (NLL) and expected calibration errors (ECE) for different corruption intensities are plotted in Fig. 4. LIVI retains a low NLL across all corruption intensities, showing that its predictive uncertainty is better calibrated than competitors. Similarly, LIVI achieves consistently low ECEs, showing that its accuracy and confidence are well-aligned (i.e., decrease at a similar rate), even as the corruption intensity increases.

## 6.6 CIFAR100

Finally, we test our approach on CIFAR100 using a WideResNet(28,10) as the BNN. Not only is this dataset more complex to model, the BNN contains roughly 36.5 million parameters, demonstrating that our approach can scale to large networks. The generator used for LIVI has only about twice

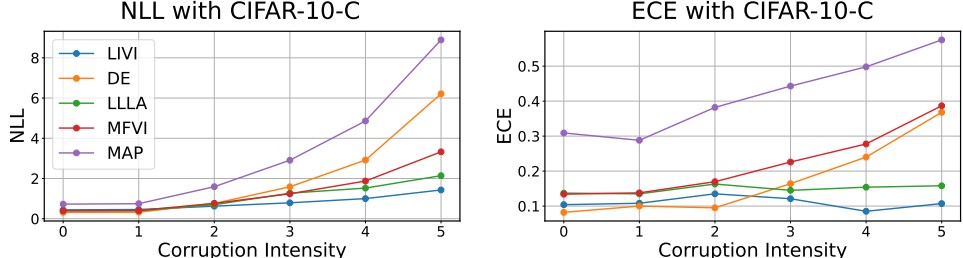

Figure 4: **OOD Test C2: Corrupted CIFAR10 benchmark.** OOD performance for methods trained on CIFAR10 and making predictions for CIFAR-10-C images corrupted with Gaussian blur (Hendrycks et al., 2019). LIVI performs as well or better than competitors.

Table 3: **CIFAR100 results.** Posterior testing with a WideResNet(28,10) architecture on the CIFAR100 dataset. The numbers reported here are on the test set, with those for LIVI being averaged across three seeds.

| Method | Accuracy (%) ↑ | NLL ↓ | ECE ↓ |
|---|---|---|---|
| MAP[†] | 79.8 | 0.875 | 0.086 |
| MFVI[†] | 77.8 | 0.944 | 0.097 |
| LIVI | 78.7 ± 0.4 | 0.774 ± 0.080 | 0.036 ± 0.004 |
| Ensemble $(n = 4)$[†] | **82.7** | **0.666** | **0.021** |

[†] Results from Nado et al. (2021).

as many (variational) parameters as the BNN latent variables – the same amount of variational parameters required for a mean-field Gaussian posterior.

We compare our results with those presented by Nado et al. (2021), who use the same BNN architecture, see Table 3. This architecture is also used by Sharma et al. (2023), who test performance on this dataset in various sub-stochastic settings and report on the same metrics, with which our results can therefore also be compared. The results indicate that LIVI's learned posterior approaches the performance of an ensemble with a combined 146 million parameters, nearly twice the roughly 75 million parameters required by LIVI. Moreover, LIVI's posterior approximation appears better than a simpler approximation like MFVI, even though this uses the same amount of parameters.

# 7   Conclusion

In this paper, we present a novel method to scale implicit variational inference to latent variable models with millions of dimensions by circumventing the need for a discriminator network and an adversarial loss. We find that modelling the posterior with a highly flexible approximation indeed does have benefits in the OOD and data drift scenarios, which may be due to the approximate posterior capturing richer information like correlations and multiple modes.   Our method performs better than deep ensembles in a number of OOD benchmarks. Unlike conventional probabilistic methods, we do not fall short on in-distribution accuracy. We empirically evaluated our bound on Bayesian neural networks with millions of latent variables. Our method can also be applied for inference in other latent variable models, such as VAEs (Kingma et al., 2014; Rezende et al., 2014; Mescheder et al., 2017a). Future work includes scaling our methods to models with hundreds of millions of parameters. For this, one could consider independent hypernetworks for each target model layer, thus reducing computational requirements at the cost of losing correlations across the layers. One could also consider efficient ways of putting priors over deep networks to curb dimensionality, such as the works by Karaletsos et al. (2018) and Trinh et al. (2020).

# 8   Acknowledgements

The authors thank Søren Hauberg and Pierre-Alexandre Mattei for providing their insight multiple times during the development of this project. We acknowledge EuroHPC Joint Undertaking for

awarding us access to Karolina at IT4Innovations, Czech Republic. AU, WB, and JF acknowledge funding from the Novo Nordisk Foundation through the Centre for Basic Machine Learning Research in Life Science (MLLS, NNF20OC0062606). KSS acknowledges funding from the Novo Nordisk Foundation (NNF20OC0065611). Furthermore, JF was in part funded by the Novo Nordisk Foundation (NNF20OC0065611), the Innovation Fund Denmark (0175-00014B) and Independent Research Fund Denmark (9131-00082B).

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

# Implicit Variational Inference for High-Dimensional Posteriors

## Supplementary material

## A  Derivation of the entropy approximation

In this section, we show that the approximation of the differential entropy in Eq. (13) can be further approximated by Eq. (14). Starting from Eq. (13), the entropy of a Gaussian DLVM can be approximated by $\tilde{H}[\tilde{q}(\boldsymbol{\theta})]$, that is

$$H[q_{\boldsymbol{\gamma}}(\boldsymbol{\theta})] = -\mathbb{E}_{\boldsymbol{z} \sim q(\boldsymbol{z})}\mathbb{E}_{\boldsymbol{\theta} \sim q_{\boldsymbol{\gamma}}(\boldsymbol{\theta} \mid \boldsymbol{z})}[\log q_{\boldsymbol{\gamma}}(\boldsymbol{\theta})] \approx \underbrace{-\mathbb{E}_{\boldsymbol{z} \sim q(\boldsymbol{z})}\mathbb{E}_{\boldsymbol{\theta} \sim q_{\boldsymbol{\gamma}}(\boldsymbol{\theta} \mid \boldsymbol{z})}[\log \tilde{q}_{\boldsymbol{z}'=\boldsymbol{z}}(\boldsymbol{\theta})]}_{=:\tilde{H}[\tilde{q}(\boldsymbol{\theta})]}, \quad (\text{A.1})$$

where we use the subscript $\boldsymbol{z}' = \boldsymbol{z}$ to specify that $\tilde{q}$ is a linearised distribution that is linearised at $\boldsymbol{z}$ and distinct for each sample from the base distribution $q(\boldsymbol{z})$. Using the definition of $\tilde{q}$ from Eqs. (11) and (12), we have

$$\log \tilde{q}_{\boldsymbol{z}'}(\boldsymbol{\theta}) = -\frac{m}{2}\log 2\pi - \frac{1}{2}\log \det \boldsymbol{C}(\boldsymbol{z}') - \underbrace{\frac{1}{2}(\boldsymbol{\theta} - \boldsymbol{\mu}(\boldsymbol{z}'))^{\mathsf{T}}\boldsymbol{C}(\boldsymbol{z}')^{-1}(\boldsymbol{\theta} - \boldsymbol{\mu}(\boldsymbol{z}'))}_{=:h(\boldsymbol{\theta}, \boldsymbol{z}')}, \quad (\text{A.2})$$

where, c.f. Eq. (12),

$$\boldsymbol{\mu}_{\boldsymbol{\gamma}}(\boldsymbol{z}') = g_{\boldsymbol{\gamma}}(\boldsymbol{z}') - \boldsymbol{J}_g(\boldsymbol{z}')\,\boldsymbol{z}', \qquad \boldsymbol{C}(\boldsymbol{z}') = \boldsymbol{J}_g(\boldsymbol{z}')\boldsymbol{J}_g(\boldsymbol{z}')^{\mathsf{T}} + \sigma^2 \boldsymbol{I}_m. \quad (\text{A.3})$$

Our approximation of the entropy is thus

$$\tilde{H}[\tilde{q}(\boldsymbol{\theta})] = \frac{m}{2}\log 2\pi + \frac{1}{2}\mathbb{E}_{\boldsymbol{z} \sim q(\boldsymbol{z})}[\log \det \boldsymbol{C}(\boldsymbol{z})] + \mathbb{E}_{\boldsymbol{z} \sim q(\boldsymbol{z})}\mathbb{E}_{\boldsymbol{\theta} \sim q_{\boldsymbol{\gamma}}(\boldsymbol{\theta} \mid \boldsymbol{z})}[h(\boldsymbol{\theta}, \boldsymbol{z})]. \quad (\text{A.4})$$

We rewrite the last term of Eq. (A.4) by using Eq. (380) from Petersen et al. (2012) to solve the inner expectation obtaining

$$\mathbb{E}_{\boldsymbol{z} \sim q(\boldsymbol{z})}\mathbb{E}_{\boldsymbol{\theta} \sim q_{\boldsymbol{\gamma}}(\boldsymbol{\theta} \mid \boldsymbol{z})}[h(\boldsymbol{\theta}, \boldsymbol{z})] = \mathbb{E}_{\boldsymbol{z} \sim q(\boldsymbol{z})}\mathbb{E}_{\boldsymbol{\theta} \sim q_{\boldsymbol{\gamma}}(\boldsymbol{\theta} \mid \boldsymbol{z})}\left[\frac{1}{2}(\boldsymbol{\theta} - \boldsymbol{\mu}(\boldsymbol{z}))^{\mathsf{T}}\boldsymbol{C}(\boldsymbol{z})^{-1}(\boldsymbol{\theta} - \boldsymbol{\mu}(\boldsymbol{z}))\right] \quad (\text{A.5})$$

$$= \frac{1}{2}\mathbb{E}_{\boldsymbol{z} \sim q(\boldsymbol{z})}\left[(g_{\boldsymbol{\gamma}}(\boldsymbol{z}) - \boldsymbol{\mu}(\boldsymbol{z}))^{\mathsf{T}}\boldsymbol{C}(\boldsymbol{z})^{-1}(g_{\boldsymbol{\gamma}}(\boldsymbol{z}) - \boldsymbol{\mu}(\boldsymbol{z})) + \mathrm{tr}(\boldsymbol{C}(\boldsymbol{z})^{-1}\sigma^2 \boldsymbol{I}_m)\right] \quad (\text{A.6})$$

$$= \frac{1}{2}\mathbb{E}_{\boldsymbol{z} \sim q(\boldsymbol{z})}\left[(\boldsymbol{J}_g(\boldsymbol{z})\boldsymbol{z})^{\mathsf{T}}\boldsymbol{C}(\boldsymbol{z})^{-1}(\boldsymbol{J}_g(\boldsymbol{z})\boldsymbol{z}) + \mathrm{tr}(\boldsymbol{C}(\boldsymbol{z})^{-1}\sigma^2 \boldsymbol{I}_m)\right], \quad (\text{A.7})$$

where, in Eq. (A.7), we substituted $\boldsymbol{\mu}(\boldsymbol{z})$ by $g_{\boldsymbol{\gamma}}(\boldsymbol{z}) - \boldsymbol{J}_g(\boldsymbol{z})\,\boldsymbol{z}$.

Now, we rewrite the two terms inside the expectation of Eq. (A.7) and apply the limit $\sigma \to 0$, following our small noise assumption in Section 3. For the first term inside the expectation, we get

$$(\boldsymbol{J}_g(\boldsymbol{z})\boldsymbol{z})^{\mathsf{T}}\boldsymbol{C}(\boldsymbol{z})^{-1}(\boldsymbol{J}_g(\boldsymbol{z})\boldsymbol{z}) = (\boldsymbol{J}_g(\boldsymbol{z})\boldsymbol{z})^{\mathsf{T}}\left(\boldsymbol{J}_g(\boldsymbol{z})\boldsymbol{J}_g^{\mathsf{T}}(\boldsymbol{z}) + \sigma^2 \boldsymbol{I}_m\right)^{-1}(\boldsymbol{J}_g(\boldsymbol{z})\boldsymbol{z}) \quad (\text{A.8})$$

$$= (\boldsymbol{J}_g(\boldsymbol{z})\boldsymbol{z})^{\mathsf{T}}\sigma^{-2}\left(\boldsymbol{J}_g(\boldsymbol{z})\boldsymbol{J}_g^{\mathsf{T}}(\boldsymbol{z})\sigma^{-2} + \boldsymbol{I}_m\right)^{-1}\boldsymbol{J}_g(\boldsymbol{z})\boldsymbol{z} \quad (\text{A.9})$$

$$= \sigma^2 \boldsymbol{z}^{\mathsf{T}}\left(\sigma^{-2}\boldsymbol{I}_d \boldsymbol{J}_g^{\mathsf{T}}(\boldsymbol{z})\left(\boldsymbol{J}_g(\boldsymbol{z})\sigma^{-2}\boldsymbol{I}_d \boldsymbol{J}_g^{\mathsf{T}}(\boldsymbol{z}) + \boldsymbol{I}_m\right)^{-1}\boldsymbol{J}_g(\boldsymbol{z})\sigma^{-2}\boldsymbol{I}_d\right)\boldsymbol{z} \quad (\text{A.10})$$

$$= \sigma^2 \boldsymbol{z}^{\mathsf{T}}\left(\sigma^{-2}\boldsymbol{I}_d - \left(\sigma^2 \boldsymbol{I}_d + \boldsymbol{J}_g^{\mathsf{T}}(\boldsymbol{z})\boldsymbol{J}_g(\boldsymbol{z})\right)^{-1}\right)\boldsymbol{z} \quad (\text{A.11})$$

$$= \boldsymbol{z}^{\mathsf{T}}\left(\boldsymbol{I}_d - \sigma^2\left(\sigma^2 \boldsymbol{I}_d + \boldsymbol{J}_g^{\mathsf{T}}(\boldsymbol{z})\boldsymbol{J}_g(\boldsymbol{z})\right)^{-1}\right)\boldsymbol{z}, \quad (\text{A.12})$$

where we obtained Eq. (A.11) using Eq. (159) by Petersen et al. (2012). Now, taking the limit of Eq. (A.12) as $\sigma \to 0$, we obtain

$$\lim_{\sigma \to 0} \boldsymbol{z}^{\mathsf{T}} \left( \boldsymbol{I}_d - \sigma^2 \left( \sigma^2 \boldsymbol{I}_d + \boldsymbol{J}_g^{\mathsf{T}}(\boldsymbol{z}) \boldsymbol{J}_g(\boldsymbol{z}) \right)^{-1} \right) \boldsymbol{z} = \boldsymbol{z}^{\mathsf{T}} \boldsymbol{z}. \tag{A.13}$$

For the second term in Eq. (A.7), we use some basic identities for matrix traces and inverse. Namely, a constant term inside a trace can be moved outside, and the trace of a matrix is the sum of its eigenvalues (Petersen et al., 2012, Eq. 12). Furthermore, eigenvalues are reciprocated when the corresponding matrix is inverted (Petersen et al., 2012, Eq. 287). We have that

$$\operatorname{tr}(\boldsymbol{C}(\boldsymbol{z})^{-1} \sigma^2 \boldsymbol{I}_m) = \sigma^2 \operatorname{tr} \left( (\boldsymbol{J}_g(\boldsymbol{z}) \boldsymbol{J}_g^{\mathsf{T}}(\boldsymbol{z}) + \sigma^2 \boldsymbol{I}_m)^{-1} \right) \tag{A.14}$$

$$= \sigma^2 \left[ \sum_{i=1}^{d} \frac{1}{s_i^2 + \sigma^2} + \sum_{d+1}^{m} \frac{1}{\sigma^2} \right] = \sum_{i=1}^{d} \frac{\sigma^2}{s_i^2 + \sigma^2} + \sum_{d+1}^{m} \frac{\sigma^2}{\sigma^2} \tag{A.15}$$

$$= (m - d) + \sum_{i=1}^{d} \frac{\sigma^2}{s_i^2 + \sigma^2}. \tag{A.16}$$

In Eq. (A.15), we use the squared singular values of $\boldsymbol{J}_g(\boldsymbol{z})$ as per singular value decomposition (Petersen et al., 2012, Eq. 292). As before, we consider the limit of Eq. (A.16) as $\sigma \to 0$ and obtain

$$\lim_{\sigma \to 0} \operatorname{tr}(\boldsymbol{C}(\boldsymbol{z})^{-1} \sigma^2 \boldsymbol{I}_m) = m - d. \tag{A.17}$$

For small values of the output variance $\sigma^2$, we approximate the last term of $\tilde{H}[\tilde{q}(\boldsymbol{\theta})]$ in Eq. (A.4) by its limit as $\sigma \to 0$. Using the limits from Eqs. (A.13) and (A.17), we have

$$\mathbb{E}_{\boldsymbol{z} \sim q(\boldsymbol{z})} \mathbb{E}_{\boldsymbol{\theta} \sim q_{\boldsymbol{\gamma}}(\boldsymbol{\theta} \,|\, \boldsymbol{z})}[h(\boldsymbol{\theta}, \boldsymbol{z})] \approx \lim_{\sigma \to 0} \mathbb{E}_{\boldsymbol{z} \sim q(\boldsymbol{z})} \mathbb{E}_{\boldsymbol{\theta} \sim q_{\boldsymbol{\gamma}}(\boldsymbol{\theta} \,|\, \boldsymbol{z})}[h(\boldsymbol{\theta}, \boldsymbol{z})] \tag{A.18}$$

$$= \frac{1}{2} \mathbb{E}_{\boldsymbol{z} \sim q(\boldsymbol{z})} \left[ \boldsymbol{z}^{\mathsf{T}} \boldsymbol{z} + m - d \right] \tag{A.19}$$

$$= \frac{\mathbb{E}_{\boldsymbol{z} \sim q(\boldsymbol{z})} \left[ \boldsymbol{z}^{\mathsf{T}} \boldsymbol{z} \right] + m - d}{2} = \frac{d + m - d}{2} = \frac{m}{2}. \tag{A.20}$$

Applying the approximation in Eq. (A.20) to Eq. (A.4), gives us the final approximation

$$\tilde{H}[\tilde{q}(\boldsymbol{\theta})] \approx \frac{m}{2} + \frac{m}{2} \log 2\pi + \frac{1}{2} \mathbb{E}_{\boldsymbol{z} \sim q(\boldsymbol{z})} \left[ \log \det \left( \boldsymbol{J}_g(\boldsymbol{z}) \boldsymbol{J}_g(\boldsymbol{z})^{\mathsf{T}} + \sigma^2 \boldsymbol{I}_m \right) \right], \tag{A.21}$$

which is also stated in Eq. (14). It is important to note here that every Jacobian (w.r.t. input $\boldsymbol{z}$) matrix for generator map $g_{\boldsymbol{\gamma}}$ is a function of $\boldsymbol{\gamma}$, the variational parameters, and hence needs to be differentiable. To obtain whole Jacobian matrices, we use the `nnj` wrapper from the StochMan library (Detlefsen et al., 2021), which gives us Jacobians in a forward pass rather than requiring a for loop over backward passes.

## B  Reparameterisation of the LIVI bounds

We note that it is trivial to reparameterise the LIVI bounds as the Gaussian DLVM is straightforward to reparameterise. Using the base variables $\boldsymbol{z}, \boldsymbol{\eta}$, the generative process for a Gaussian DLVM, c.f. Appendix D, can be reparameterise as

$$\boldsymbol{\theta}' = g_{\boldsymbol{\gamma}}(\boldsymbol{z}), \quad \boldsymbol{z} \sim \mathcal{N}(0, \boldsymbol{I}_d) \tag{B.1}$$

$$\boldsymbol{\theta} = \boldsymbol{\theta}' + \boldsymbol{\eta}, \quad \boldsymbol{\eta} \sim \mathcal{N}(0, \sigma^2 \boldsymbol{I}_m). \tag{B.2}$$

The reparameterised LIVI with whole Jacobians, corresponding to Eq. (15), is

$$\mathcal{L}'(\gamma) = \mathbb{E}_{\boldsymbol{z} \sim q(\boldsymbol{z}), \boldsymbol{\eta} \sim q(\boldsymbol{\eta})} \left[ \log p(\mathcal{D} \,|\, g_{\boldsymbol{\gamma}}(\boldsymbol{z}) + \boldsymbol{\eta}) + \log p(g_{\boldsymbol{\gamma}}(\boldsymbol{z}) + \boldsymbol{\eta}) \right.$$
$$\left. + \frac{1}{2} \log \det \left( \boldsymbol{J}_g(\boldsymbol{z}) \boldsymbol{J}_g(\boldsymbol{z})^{\mathsf{T}} + \sigma^2 \boldsymbol{I}_m \right) \right] + c, \tag{B.3}$$

and the reparameterised LIVI bound with a differentiable lower bound on the determinant, corresponding to Eq. (18), is

$$\mathcal{L}''(\gamma) = \mathbb{E}_{\boldsymbol{z} \sim q(\boldsymbol{z}), \boldsymbol{\eta} \sim q(\boldsymbol{\eta})} \left[ \log p(\mathcal{D} \mid g_{\boldsymbol{\gamma}}(\boldsymbol{z}) + \boldsymbol{\eta}) + \log p(g_{\boldsymbol{\gamma}}(\boldsymbol{z}) + \boldsymbol{\eta}) + \frac{d}{2} \log(s_1^2(\boldsymbol{z}) + \sigma^2) \right] + c. \tag{B.4}$$

## C    Singular values explained

Here we write out the steps that lead from the entropy approximation in Eq. (15) to the efficient entropy approximation we actually use Eq. (16). The first step is to note that the determinant of a square matrix can be written as the product of its eigenvalues and then the log determinant by extension can be written as the sum of the log of these eigenvalues. Let $s_1(\boldsymbol{z}), \ldots, s_m(\boldsymbol{z})$ be the singular values of the Jacobian $\boldsymbol{J}_g(\boldsymbol{z})^\intercal$ (including zeros), using Petersen et al. (Eq. 286, 2012) we can write the log determinant term in Eq. (15) as

$$\det(\boldsymbol{J}_g \boldsymbol{J}_g^\intercal) = \prod_{i=1}^{m} s_i^2(\boldsymbol{z}) \tag{C.1}$$

Strictly speaking, the matrix $\boldsymbol{J}_g \boldsymbol{J}_g^\intercal$ is not full rank and has only $d$ non-zero singular values, and thus this determinant will be equal to 0. The additional noise added at the output of the generator resolves this issue as it appears in the $\boldsymbol{C}(\boldsymbol{z})$ in Eq. (A.1).

$$\det(\boldsymbol{J}_g \boldsymbol{J}_g^\intercal + \sigma^2 \boldsymbol{I}_m) = \prod_{i=1}^{d} \left[ s_i^2(\boldsymbol{z}) + \sigma^2 \right] \sigma^{2(m-d)} \tag{C.2}$$

$$\log \det(\boldsymbol{J}_g \boldsymbol{J}_g^\intercal + \sigma^2 \boldsymbol{I}_m) = \sum_{i=1}^{d} \log \left[ s_i^2(\boldsymbol{z}) + \sigma^2 \right] + 2(m-d) \log \sigma \tag{C.3}$$

As we do not wish to calculate all the singular values to estimate Eq. (C.3) exactly, because this would be computationally expensive, we resort to using a lower bound in Eq. (17). The terms in the summation in Eq. (C.3) are positive, so estimating this with any one singular value should give us a lower bound.

Depending on what singular value we wish to compute, we can use the LOBPCG algorithm, which iteratively optimises the generalised Rayleigh quotient and converges in linear time. All details about the convergence results are available in Knyazev (2001). We also employ an efficient backward mode to get vector-jacobian products to work with the generalised Rayleigh quotient, and thus we never need to store large Jacobian matrices during the calculation of this singular value. The Rayleigh quotient is given as follows:

$$\rho(\boldsymbol{v}) = \frac{\boldsymbol{v}^\intercal \boldsymbol{J}_g^\intercal \boldsymbol{J}_g \boldsymbol{v}}{\boldsymbol{v}^\intercal \boldsymbol{v}} \tag{C.4}$$

## D    Novel generator architecture

Here we describe the novel generator architecture. A central property is that we drop dense connections deeper into the network. This resulted in an architecture that had fully connected matrix multiplication layers at the start and smaller individual networks deeper into the architecture. In other words, to decrease the number of parameters closer to the output of the network, dense connections are pruned away. The final output vector of parameters is generated via the concatenation of outputs from smaller networks, which can be thought of as producing weights for each layer in the BNN.

The architecture is pictorially represented in Fig. D.1. All computational blocks, as well as the input block in this network, are a matrices, as we follow the standard recipe of the Matrix multiplication neural network (Shi et al., 2018). Starting from the left, we have a large noise vector represented in matrix format. We use the 65x65-dimensional input noise matrix for MNIST experiments and the 130x130 input noise matrix for the CIFAR10 experiments. The noise is sampled from the standard normal distribution. Then we use a matrix multiplication layer to produce a higher dimensional output that is split and passed on to a series of disconnected individual matrix multiplication networks

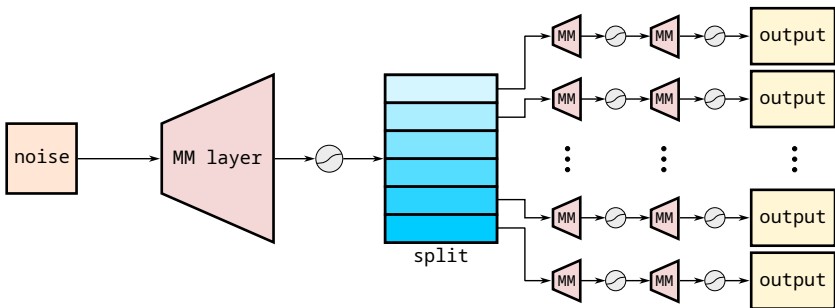

Figure D.1: Our novel `CMMNN` architecture.

Table E.1: **Dual Networks Setups.** With this table we clarify the architectures we use for the primary (hypernetwork) and the secondary (BNN) network for training with all the different datasets used for experiments.

|  | Toy Data | UCI Reg | MNIST | CIFAR10 | CIFAR100 |
|---|---|---|---|---|---|
| Hypernetwork | MLP | MLP | MMNN | CMMNN | CMMNN |
| BNN | MLP | MLP | LeNet | WideResNet(16,4) | WideResNet(28,10) |

that produce weights and biases for individual layers of the WideResNet BNN. This way the task of capturing correlations within layers is taken care of by the individual sub-networks towards the end and capturing correlations across layers is taken care of by the homogeneous layer at the start of the network.

# E    Experiment Details

## E.1    Metrics

For regression, we test generalisation by measuring negative log-likelihoods and mean squared error over the test set.

For the classification experiments, we measure the averaged OOD AUROC and confidence (maximum softmax value) (Daxberger et al., 2021a) on OOD datasets. The OOD AUROC and confidence are crucial metrics to understand the behaviour of the BNN on unseen data. A low confidence score indicates that the model is very uncertain about a classification, while a high AUROC score suggests that the model does not misclassify unseen data with high probability or confidence. Such misclassifications would increase the false positive rate, thereby lowering the AUROC. Therefore, these metrics allow us to measure the model's generalisation performance on OOD data hinting at the quality of the posterior. We further consider the negative log-likelihood (NLL) and expected calibration error (ECE) for some OOD classification experiments. These two metrics ensure that the implicit approximation is not placing mass in undesirable regions of the posterior and is learning the right variance in the function space. If the NLL is lower for OOD points, it signifies that the functional variance farther from the data increases, which is desirable. At the same time, a low ECE indicates that the model's predictive confidence falls with its accuracy as per expectations.

## E.2    Experimental setup

Details of the architectures used for the generator/hypernetwork and the Bayesian neural network for all of the experiments in Section 6, can be found in Table E.1. For reference, our novel CMMNN architecture is implemented as `CorreMMGenerator` in our code.

## E.3    Toy experiments

We use a 2D toy sinusoidal dataset for this example with 70 training data points. We intentionally leave a gap in the middle to visualise the epistemic uncertainty or the diversity of functions in this

region as this has been reported by Foong et al. (2019) to be tricky or unachievable for simpler variational approximations.

The BNN architecture is shared across all the methods and is a two-hidden-layered MLP with 7 and 10 units and ELU activations trained with losses unique to each of the methods. We also assume homoscedastic noise in the data that is estimated by all the methods using an extra parameter learnt via type II maximum likelihood. This is set in the data generating process at 0.3.

**LIVI:** Here, a generator network, which is a one-hidden layered MLP with an 80-dimensional noise input and ELU activations, produces a 105-dimensional output vector which is then used to reparameterise the layers (weights and biases) of a BNN for training.

**MFVI:** For mean-field VI, we take advantage of the `bayesian-torch` library (Krishnan et al., 2022). We construct the exact same architecture for the BNN using the reparameterisable layers provided by this library and we slowly warm up the KL term reaching 1 towards the end of training. This is done to help the method converge.

**HMC:** We sample 10000 samples after a warmup of 2000 samples using the same and neural network architecture. We use the NUTS sampler available in the `Pyro` (Bingham et al., 2018) package.

**AVB:** We also compare our method qualitatively against the conventional style of training implicit distributions which uses a density ratio estimating discriminator network. Using the same architecture as above for the BNN, we use a one-hidden layered discriminator network that we retrain within each training loop of the BNN to discriminate 20 samples from the prior and the posterior (generator) each. We run this inner training loop for 10 epochs. The algorithm here is exactly the same as Algorithm 1 in Mescheder et al. (2017a).

### E.4 Details of UCI experiments

We use an MLP BNN with 50 units in one hidden layer. We report the RMSE and log-likelihood on held-out data for our method. We use generator architectures with far fewer parameters than Shi et al. (2018) and do not assume independence across layers, i.e., we use one MLP to generate all the weights of the BNN. All of the generator architectures are one hidden layered MLP with a slightly varying number of units depending on the dataset.

We train our method with a homoscedastic assumption, i.e., the variance in the dataset is assumed to be constant and we train an observation noise parameter using type-II maximum likelihood.

### E.5 MNIST experiments

Of the wide-ranging approximate inference approaches we compare against, some focus on capturing multiple modes like deep ensembles, while others like MFVI capture information around a single mode sacrificing all correlations for computational feasibility. Similarly, the Laplace approximation, a popular *post hoc* method (MacKay, 1992), captures unimodal information but can be made much faster by considering a sub-stochastic Bayesian neural network (Daxberger et al., 2021a). Moreover, there is a range of approximations for estimating the hessian that again, trade-off correlations for computational feasibility. In all of these cases, a single MAP estimate remains the easiest baseline to implement and train but represents the minimum amount of information that can be obtained about the posterior surface.

For all methods, we use the same LeNet architecture and for (LLLA, DE & MAP) we train with equal weight decay of $8 \times 10^{-4}$. For MFVI, we use a standard normal prior.

**LIVI** For all our MNIST experiments we use an MMNN with $65 \times 65$ input noise, one hidden layer of size $250 \times 250$ and produce an output matrix of size $350 \times 127$, the elements of which are used as weights and biases of the LeNet BNN. For all experiments, we trained without dataset augmentation and with a maximum of 1 to 2 samples (from $q_\gamma(\theta)$) per minibatch. This network generates close to $44\,000$ parameters for the LeNet BNN while keeping its own parameters appreciably low compared to an MLP hypernetwork conserving computation required for the objective Eq. (18). The weights

and biases of the hypernetwork are the variational parameters that tune the output implicit distribution and here they number slightly over six hundred and ten thousand i.e. these are the total number of weights and biases in the hypernetwork.

**LLLA**   The full Hessian for the last layer can be obtained very cheaply as the last layer Hessian is simply the *Generalised Gauss Newton* matrix since the network is linear in the last-layer weights (Immer et al., 2021). Hence, we strike a balance between expressivity and computational cost by obtaining the most accurate *post-hoc* covariance structure over a *sub-stochastic network*. We use a pretrained LeNet with a weight decay of 8e-4 for this method.

**DE**   To represent deep ensembles we use five LeNet networks trained with different seeds and average over their predictions and use one of these networks to represent the MAP estimate. Each network is trained with a weight decay of 8e-4.

**MFVI**   We use weights from a pre-trained LeNet for initialising the mean and retrain it for several more epochs using the ELBO and a standard normal prior to obtain an isotropic Gaussian posterior. Again this is done using `bayesian-torch` library and the MOPED functionality (Krishnan et al., 2020).

**AVB**   We again compare our method to the existing adversarial training routine for implicit distributions which is represented by AVB (Algorithm 1, Mescheder et al., 2017a). For this method, we use an MLP discriminator with two hidden layers containing 2500 and 4500 neurons for classifying if a sample comes from the generator or the prior (regularisation term in Eq. (2)). We use 300 samples from the prior and the posterior (generator) for a full batch training of the discriminator for 5 epochs within each minibatch update of the main training loop.

**KIVI**   To compare our method against Shi et al. (2018) we also use MLP BNNs as per their experimental setup and outperform on in-distribution test accuracy reported in Shi et al. (2018, Figure 2, left table). To test fairly against KIVI we use MLP BNN architectures only for this experiment as a LeNet would've been easily able to achieve better accuracy on the in-distribution test set. We find that with the same BNN and hypernetwork architectures we get better test accuracies on MNIST

**Training**   To train LIVI, we use a Matrix Multiplication Neural Network (MMNN, Shi et al., 2018) as the hypernetwork architecture. For the LLLA, we use a complete ("full", Daxberger et al., 2021a) Hessian for the weights and biases in the last layer obtaining all correlations over a smaller parameter set to ensure feasibility. For deep ensembles, we average over five networks trained with weight decay. For one-to-one comparison with AVB we use the exact same generator and training hyperparameters as LIVI. To train LIVI with the aforementioned generator architecture it takes 12.3 GPU minutes on a Titax X (Pacal 12gb vram) GPU and consumes 980 megabytes of memory when training with a batch size of 256. On the other hand, to train a LeNet deep neural network with the same batch size on the same GPU it takes 3.07 GPU minutes and 700 megabytes of GPU memory.

### E.6   CIFAR10 Experiments

**LIVI**   We use our novel generator architecture for this experiment with a noise dimension of 130x130. This architecture and its configuration file for matrix multiplication layers are present in the code. We train with a batch size of 256 and with 2 samples per minibatch. We use Eq. (17) elbo as the objective for training. Our generator architecture produces all the parameters of the BNN, implicitly learning the correlation between and across layers. This hypernetwork contains about 7 million parameters, in other words, these are the number of variational parameters that tune the implicit distribution.

**DE**   We train five WideResNet (Zagoruyko et al., 2016), each 16 layers deep with different initialisations using the same optimiser hyperparameters i.e. with NAdam optimiser with weight decay of $1 \times 10^{-4}$, a learning rate of $7.8 \times 10^{-4}$ and with cosine annealing of the learning rate. We average predictions from these 5 networks to represent deep ensembles in our experiments.

**MAP**   We train one WideResNet with the same weight decay and optimiser hyperparameters as above and use that for representing the MAP estimate.

**LLLA**  For the last-layer Laplace approximation, we use a network trained similarly to a MAP network and then run post hoc Laplace on it, inferring a full Hessian matrix for all of the last-layer weights. Again this is done to strike a middle ground between inferring a very faithful posterior and keeping the computational requirements in check.

**AVB**  Even though AVB is theoretically very close to our variational approximation, the size of the discriminator to distinguish 2.7 million dimensional samples from the prior and the implicit posterior to estimate the regularisation term in Eq. (2) poses a prohibitively large overhead. Due to this, we could not train AVB for this set of experiments.

**Training**  Interestingly, our method trained using Eq. (16) takes less time to train than a deep ensemble consisting of five networks with the same training parameters and provides better uncertainty quantification than a non-Bayesian ensemble of neural networks. Keeping everything else the same, each network of the ensemble takes 34 minutes to train on CIFAR10 using an A5000(24 gb vram) for a total of 2.83 GPU hours whilst our method/architecture requires 2.30 GPU hours to train on the same dataset using the same GPU. We use Eq. (17) to train our implicit approximation and consume 4.1 GB of memory with a batch size of 256 and training with 2 elbo samples per minibatch. A vanilla WideResNet (MAP)consumes 1.4 GB to train with the same batch size.

## E.7   Details of Fig. 1

In Fig. 1, we compare both the objective functions presented in this work for training with implicit variational approximations to different methods for uncertainty quantification for neural networks. All models were trained for five thousand iterations and had to learn observation noise present in the toy sinusoidal dataset. For all methods, we train with a homoscedastic assumption and obtain an observation noise parameter using type-II maximum likelihood. We deliberately removed a part of the data to see if the models tested were able to find in-between uncertainties. All methods were given the same sized 2 hidden layered MLP with 7 and 10 units respectively. We trained 5 MAP networks with different seeds for Deep Ensembles and average their predictions to make the plot. The variance of the predictions was then used for the epistemic uncertainty in blue. We also train the model with an observation noise parameter. For MFVI, we annealed the KL divergence between the prior and the posterior with a weighting term to help it to converge. The weighting term reaches one as the training nears completion. For HMC we sample 10000 samples using the library `Pyro`. We also tried to make multiplicative normalizing flows converge for this dataset, but with even 20K parameters and training for ten thousand iterations with a very small learning rate did not help. We even tried KL down weighting to reduce the effect of the prior in the initial iterations but that did not work either.

After training, we also plot a KDE-plot of the samples from the generator in Appendix F.4. As a sanity check, we do confirm that the generator is capable of representing non-trivial distributions as we can spot heavy tails and multiple modes in these plots. In this experiment, we qualitatively compare the epistemic uncertainty resulting from the respective inference schemes in the data-scarce regions.

## E.8   Computation Graph

Here we provide some details about how the combination of the joint generator-BNN model works. The Bayesian neural network classes for all types of architectures(feed-forward, convolutions, etc.) require a generator in the `init` function. As such, the generator networks reside inside the BNN and reparametrise it with a simple `sample_parameters` function. The most important part of this kind of implementation was the layers themselves. **PyTorch** provides different kinds of mutable layer implementations in `nn.module` but these layers do not expose their state i.e. their parameters in a manner that allows changes on the fly during training. We reimplemented the layers allowing such resampling to occur with the generator. In the `init` function of the BNN, we generate one set of parameters with the generator, package it in a `dict` that has the weight sample as well as a index to know the number of weights used by a previous layer. This counter index is updated by each layer in their `init` and `sample_parameters` function. As such, only the parameters of the generator are trainable, the parameters of the BNN are switchable and relay gradients to the generator via the likelihood or the entropy term.

Table F.1: **UCI regression datasets.** We report RMSE (↓) on the test set and average across three different seeds for each model to quantify the variance in the results.

| Method | Boston | Concrete | Energy | Kin8nm | Naval |
|---|---|---|---|---|---|
| LIVI ($\mathcal{L}'$) | 2.32 ± 0.07 | **4.24 ± 0.17** | 0.41 ± 0.27 | **0.03 ± 0.00** | **0.00 ± 0.00** |
| LIVI ($\mathcal{L}''$) | 2.40 ± 0.09 | 4.62 ± 0.13 | 0.44 ± 0.11 | 0.08 ± 0.01 | 0.00 ± 0.01 |
| HMC | **2.26 ± 0.00** | 4.27 ± 0.00 | **0.38 ± 0.00** | 0.04 ± 0.00 | **0.00 ± 0.00** |
| DE | 3.28 ± 1.00 | 6.03 ± 0.58 | 2.09 ± 0.29 | 0.09 ± 0.00 | 0.00 ± 0.00 |
| KIVI | 2.80 ± 0.17 | 4.70 ± 0.12 | 0.47 ± 0.02 | 0.08 ± 0.00 | 0.00 ± 0.00 |
| MNF | 3.31 ± 0.10 | 5.82 ± 0.04 | 1.04 ± 0.01 | 0.08 ± 0.01 | 0.01 ± 0.00 |

Table F.2: **UCI regression datasets.** We report log-likelihood (↑) on the test set and average across three different seeds for each model to quantify the variance in the results.

| Method | Boston | Concrete | Energy | Kin8nm | Naval |
|---|---|---|---|---|---|
| LIVI ($\mathcal{L}'$) | **−2.16 ± 0.05** | −2.79 ± 0.11 | −1.17 ± 0.13 | 1.24 ± 0.04 | 6.74 ± 0.04 |
| LIVI ($\mathcal{L}''$) | −2.40 ± 0.09 | −2.99 ± 0.13 | −1.37 ± 0.11 | 1.15 ± 0.01 | 5.84 ± 0.06 |
| HMC | −2.20 ± 0.00 | **−2.67 ± 0.00** | **−1.14 ± 0.00** | 1.27 ± 0.00 | **7.79 ± 0.00** |
| DE | −2.41 ± 0.25 | −3.06 ± 0.18 | −1.31 ± 0.22 | **1.28 ± 0.02** | 5.93 ± 0.05 |
| KIVI | −2.53 ± 0.10 | −3.05 ± 0.04 | −1.30 ± 0.01 | 1.16 ± 0.01 | 5.50 ± 0.12 |
| MNF | −2.66 ± 0.08 | −3.24 ± 0.09 | −1.34 ± 0.07 | 1.10 ± 0.01 | 5.01 ± 0.00 |

# F   Additional results

In this section, we report additional results. In particular, we consider multiplicative normalising flows (MNF, Louizos et al., 2017), as well as two versions of full Laplace, 1) a non-linearised full network Laplace using a low-rank approximation to the Hessian (FLAP), and 2) a linearised full network Laplace using a Kronecker factored structure of the Generalised Gauss-Newton approximation to the Hessian (FLLA).

While we show results for linearised Laplace, as this method often achieves state-of-the-art performance, it must be emphasised that, for LIVI, we *do not* linearise the posterior; we linearise the neural sampler $g_{\gamma}(z)$, but only when it is used within the estimation the intractable entropy and its gradients, see Eqs. (10) to (12). The resulting posterior approximation is, therefore, still a highly flexible (that is, non-linearised) implicit distribution. Linearised Laplace and LIVI are, therefore, fundamentally different approaches. Whereas linearised Laplace linearises the BNN and the predictive function, we linearise the neural sampler/hyper-network within the variational distribution $q_{\gamma}(\theta \,|\, z)$, thus obtaining the approximation $\tilde{q}_{z'}(\theta \,|\, z)$, but only when it is needed to approximate the entropy. This is also the reason why we do not linearise our BNN for predictions, which is common practice with linearised Laplace.

## F.1   Additional results for UCI datasets

In Tables F.1 and F.2, we report additional results for experiments on the UCI datasets. Overall, LIVI ($\mathcal{L}'$, Eq. (15)) performs competitively, either outperforming or being on par with HMC, which is considered the gold standard. The multiplicative normalising flow (MNF) is difficult to get to converge, and both the training time and memory consumption are worse than for LIVI (see Appendix F.3).

We do not show results for full Laplace here, as we repeatedly ran into errors thrown by the second-order backends (`BackPack` and `Asdl`) of the `laplace` library (Daxberger et al., 2021a) when used regression tasks.

## F.2   Additional results for OOD experiments

We show further results for the OOD experiment Test M-C1 in Table F.3, as well as Test M2 and M3 in Figs. F.1 and F.2, respectively. Note that we were unable to scale all the methods listed in Table F.3 to work on CIFAR10. In the table and both figures, the non-linearised version of full Laplace (FLAP) underfits due to the quite crude approximation to the posterior over the full network and generates

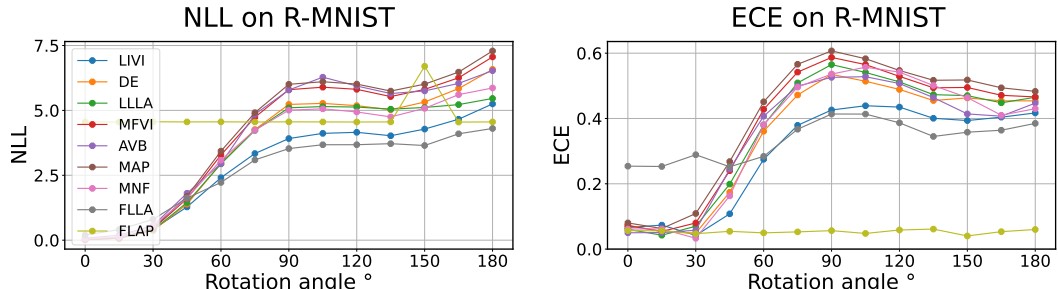

Figure F.1: Additional results for OOD Test M2: Rotated MNIST benchmark.

Table F.3: Further experiments for OOD Test M-C1. FLLA: full linearised Laplace; FLAP: full non-linearised Laplace; MNF: multiplicative normalising flows.

| | Confidence ↓ | | AUROC ↑ | |
| Method | MNIST | CIFAR10 | MNIST | CIFAR10 |
| --- | --- | --- | --- | --- |
| LLLA | 67.40 ± 0.19 | 53.6   ± 0.3 | 96.67 ± 0.27 | 89.03 ± 0.51 |
| DE | 63.14 ± 0.11 | 67.17 ± 0.21 | 97.52 ± 0.08 | 89.61 ± 0.11 |
| MAP | 72.10 ± 0.36 | 81.40 ± 0.16 | 96.32 ± 0.22 | 86.73 ± 0.64 |
| LIVI | 55.03 ± 0.13 | **43.47 ± 0.28** | **97.91 ± 0.27** | **91.83 ± 0.41** |
| AVB | 70.68 ± 0.45 | NA | 95.5   ± 0.4 | NA |
| MFVI | 69.23 ± 0.24 | 74.71 ± 0.23 | 96.53 ± 0.16 | 87.50 ± 0.25 |
| FLLA | 52.63 ± 0.17 | NA | 97.57 ± 0.19 | NA |
| FLAP | **22.40 ± 0.32** | NA | 59.55 ± 0.23 | NA |
| MNF | 61.77 ± 0.26 | NA | 96.37 ± 0.15 | NA |

poor samples. The linearised version (FLLA) works slightly better than LIVI in terms of ECE and NLL on the rotated-MNIST benchmark (Fig. F.1) and in the OOD entropy-CDF test (Fig. F.2). As noted before in Section 5 and **??**, our approach is fundamentally different from linearised Laplace approaches as we do not linearise the model during prediction and inference and hence these methods are not directly comparable.

## F.3   Runtime analysis

We report training times until convergence as well as memory consumption for LIVI and four competitors on MNIST in Table F.4. The methods were all run on the same single-GPU (NVIDIA RTX A4000) machine. Compared to other expressive VI methods like AVB and MNF, LIVI is faster to train and consumes less memory. In particular, the hypernetworks used for LIVI in all experiments only contain about twice as many parameters as the networks they are modelling the posterior over – the same amount of parameters required for mean-field VI. Interestingly, even though the memory requirements for LIVI are higher than that of DE, LIVI trains in a shorter amount of time.

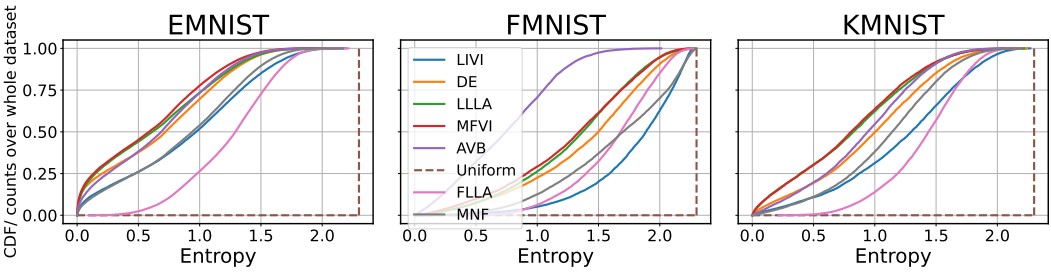

Figure F.2: Additional results for OOD Test M3: Empirical CDF plot.

Table F.4: Training times (until convergence) and memory consumptions on MNIST.

| | AVB | MFVI | LIVI | DE | MNF |
|---|---|---|---|---|---|
| Training time (minutes) | 64.1 | 10.2 | 12.3 | 15.6 | 18.7 |
| Memory (GBs) | 2.63 | 0.81 | 0.93 | 0.60 | 1.77 |

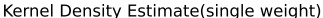

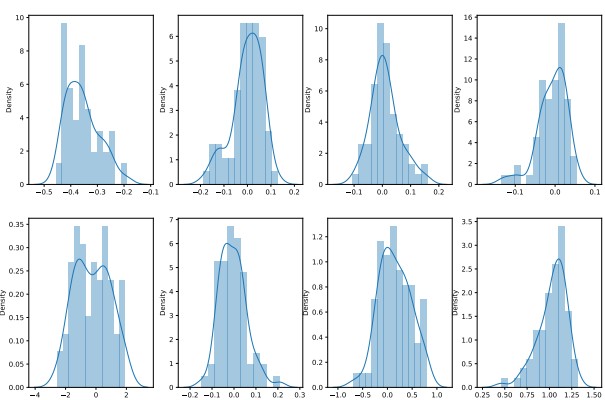

Figure F.3: Multimodal densities over weights using KDE

## F.4  KDE plot

Figure F.3 shows a KDE plot of weights randomly chosen from samples obtained from a trained generator used for the toy dataset, see section 6.2.

## F.5  Testing the entropy approximation

Because our proposal relies on approximating the entropy of an intractable variational approximation through linearisation, we test our approximation in isolation against an importance-sampled estimate of the entropy of a VAE's generative model.

$$-\mathbb{E}_{q_\gamma(\theta)}[\log q_\gamma(\theta)] = -\mathbb{E}_{q(z)}\mathbb{E}_{q_\gamma(\theta|z)}\left[\log \int_z q_\gamma(\theta|z)p(z)dz\right] \tag{F.1}$$

$$= -\mathbb{E}_{q(z)}\mathbb{E}_{q_\gamma(\theta|z)}\left[\log \mathbb{E}_{\tilde{q}(z|\theta)}\left(\frac{q_\gamma(\theta|z)q(z)}{\tilde{q}(z|\theta)}\right)\right] \tag{F.2}$$

For Fig. F.4, we estimate the entropy of a VAE decoder with a very small output variance using importance sampling as the ground truth. We choose a small output variance as this emulates our generator Section 3 which is, by construction very similar to a Gaussian VAE. Noteworthy here, the implicit variational approximation presented Section 3 does not involve an encoder architecture that parameterises the reverse distribution $\tilde{q}(z|\theta)$ and is only part of this experiment. For this analysis, the VAE was trained on HMC samples from a small BNN trained on toy sinusoidal data similar to Fig. 1. As such this VAE's decoder learns the generative distribution over parameters of a BNN very similar to the generator that parameterises the implicit variational approximation presented in this paper. The architecture of this VAE is also the same as the generator architectures used for experiments. The results here depict that even though the entropy estimated by the importance sampled ground truth and our bound in Eq. (17) are numerically different, our approximation Eq. (15) and the smallest singular value lower bound do retain and follow the characteristics of the ground truth making them viable estimators of this quantity.

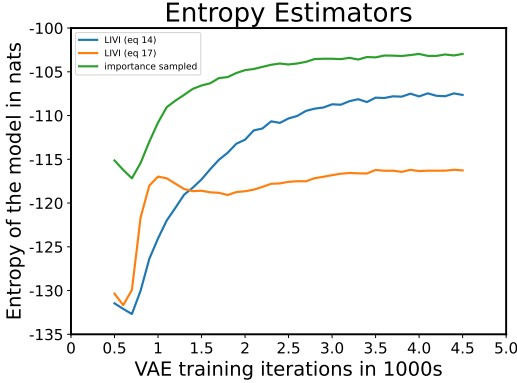

Figure F.4: Comparisons of different entropy approximations for decoder/generator through iterations. Refer to the paper for Eqs. (15) and (17) for the different entropy estimators we propose. The importance sampled estimate is presented here.

# G   Future Work

Scaling our method to models with hundreds of millions of parameters is an important line of future research. For this, one could consider independent hypernetworks for each layer of the BNN, thus reducing computational requirements at the cost of losing correlations across the layers. One could also consider efficient ways of putting priors over deep networks to curb dimensionality, such as the works by Karaletsos et al. (2018) and Trinh et al. (2020).

We also leave for the future applications of this implicit approximation to sequential learning problems like continual learning(see section 4.4 of Daxberger et al., 2021a) and also the several possible extensions of Eq. (18). We use a very simple yet potent individual singular value to approximate a $\log\det$ term but expect that this term can be better approximated minimising information loss from the Jacobian matrices without compromising computational efficiency.

