# OpenReview forum: "Implicit Variational Inference for High-Dimensional Posteriors"
_NeurIPS.cc/2023/Conference — NeurIPS 2023 spotlight_

### Official Review · Reviewer_M1uk · 2023-07-03

**Soundness:** 3 good
**Presentation:** 3 good
**Contribution:** 2 fair
**Rating:** 4
**Confidence:** 4

**Summary:**

The paper proposes LIVI a variational-inference-based approach for Bayesian NNs. It hinges on implicit VI (IVI) which allows to easily sample from an arbitrary distribution, yet obtaining the gradients w.r.t its parameters or the density is hard. As the ELBO in VI decomposes to two terms, an expected likelihood term and a KL term, under this framing the KL term cannot be evaluated, and even may be ill-defined. To remedy that, the authors propose to use a Gaussian deep latent variable model (DLVM), i.e., the generator network in VAEs, as a replacement for the implicit density. And, to approximate the entropy term in the KL divergence the authors propose to linearize the output of the network. The authors also propose a lower-bound to the ELBO by approximating the log-determinant of the covariance using the minimal singular value, which can be calculated more efficiently compared to all of the values. The authors compare their method to several Bayesian methods on in-distribution and out-of-distribution tasks showing improved performance over baseline methods.

**Strengths:**

* A novel approach for using hyper-networks to learn a Bayesian model using VI.
* Good results on OOD tasks compared to baseline methods.
* The paper is written clearly and addresses relevant related studies.
* The results seem to be reproducible - exact experiential details were given along with the code.

**Weaknesses:**

**Method**
* Although I think the idea is nice a major shortcoming of this method is that it is challenging to apply it to large networks because of the use of Hyper-networks. Even in the experimental section, the largest network has 2.7 million parameters which is considered small these days. How do the authors propose to handle that issue and scale it to networks of 1-2 orders of magnitude larger?
* To estimate the entropy term the std of the noise is taken to be zero on only one summand (the bilinear function), but not on the logdet of the covariance. How do the authors justify that?

**Experiments**
* The compared methods are somewhat outdated. Many Bayesian models were published in recent years, and all of them show that they beat deep ensembles in one way or another. I believe that more recent baselines should have been evaluated.
* I find it a bit odd that the authors didn't compare LIVI to methods with similar pipelines, such as flow-based methods [1, 2].
* The method was evaluated on UCI benchmarks, MNIST and CIFAR-10. In my opinion, it is not enough to showcase the advantage of the method and more challenging datasets should be considered.


[1] Krueger, D., Huang, C. W., Islam, R., Turner, R., Lacoste, A., & Courville, A. (2017). Bayesian hypernetworks. arXiv preprint arXiv:1710.04759.
[2] Louizos, C., & Welling, M. (2017, July). Multiplicative normalizing flows for variational bayesian neural networks. In International Conference on Machine Learning (pp. 2218-2227). PMLR.

**Questions:**

* How did your method of approximating the entropy is compared to standard MC sampling as suggested in the beginning of section 3.2?
* How does your method perform compared to baseline methods in terms of memory and run-time?

**Limitations:**

The authors did not address the limitations of their method.

---

> ### Author Rebuttal · Authors · 2023-08-09
>
> Thank you for a detailed review. We answer your concerns below.
>
> **W1: How could we make the method scale to tens or hundreds of millions of dimensions?**
> We absolutely agree that current state-of-the-art DNNs are much larger than the networks our approach can serve. However, this does not make our approach any less relevant to the Bayesian deep learning community, as research in this area does not solely focus on transformers. No prior work in the Bayesian literature has been able to scale expressive variational approximations to such a large number of latent variables.
>
> To scale our method even further, one could consider independent hypernetworks for each layer of the BNN, which would reduce the size of hypernetwork at the cost of losing correlations across layers of the BNN. This should help increase the modelling capacity significantly. One could also consider efficient ways of putting priors over deep networks to curb dimensionality like [1] that proposes implicit BNNs that have priors over activations rather than weights and biases, and [2] which also assumes priors over units in a neural network and models weights using these latent distributions.
>
> While we consider scaling to even higher dimensions as future work, we agree that the topic is important for our paper and have added a discussion on the issue of scaling to the paper.
>
> [1] Trinh et al., "Scalable Bayesian neural networks by layer-wise input augmentation", arXiv 2020, https://arxiv.org/abs/2010.13498
> [2] Karaletsos et al., "Probabilistic Meta-Representations Of Neural Networks", UDL workshop 2018, https://arxiv.org/abs/1810.00555
>
> **W2: Why is $\sigma_m^2$ not taken to be zero in the logdet of the covariance (e.g., in Eq. (A.18))?**
> It is correct that both $E_{z\sim q(z)} [\log \det C(z)]$ and $E_{z\sim q(z)} E_{\theta \sim q_\gamma(\theta | z)} [ h(\theta, z) ]$ in Eq. (A.13) depend on $\sigma^2$. For smaller networks, $E_{z\sim q(z)} [\log \det C(z)]$ can be computed analytically, but $E_{z\sim q(z)} E_{\theta \sim q_\gamma(\theta | z)} [ h(\theta, z) ]$ involves a matrix inverse, which requires a computationally expensive and unstable procedure. We, therefore, choose to approximate this term, which we do by letting $\sigma^2 \rightarrow 0$. We could have chosen other approximations, but this one makes sense, as $\sigma^2$ is already defined to be small. We do not apply the same approximation to $E_{z\sim q(z)} [\log \det C(z)]$ since we do not have to (as long as we consider small networks), and introducing an unnecessary approximation makes little sense.
>
> To derive our second bound, $\mathcal{L}''$, rather than approximating $\mathbb{E}_{z\sim q(z)} [\log \det C(z)]$, we introduce a lower bound, which was presented by Geng et al. (2021) [1]. From an optimisation perspective, a bound makes more sense than an approximation, and this bound is furthermore efficient to compute.
>
>  [1] Geng et al., "Bounds all around: training energy-based models with bidirectional bounds". NeurIPS 2021.
>
> **W3: The baselines are somewhat outdated. Many recent Bayesian models beat deep ensembles.**
> Thank you for raising this concern. We do our best to make fair comparisons by including both recent and well-performing models. We are not aware of the models you refer to, but if you have specific ones in mind, we will happily try to test them and report back to you.
>
> **W4: There should be comparisons to flow-based methods.**
> NFs are powerful models, but scaling them to millions of dimensions is incredibly difficult and computationally demanding. In fact, these challenges were some of the main motivators for our work. Regarding the multiplicative normalising flows (MNF), we have added results for these on the UCI datasets (tables 1 and 2) and MNIST (table 3), see the attached PDF. The flows are difficult to get to converge, however. Additionally, the training time and memory consumption are shown in table 4. Compared to MNF, LIVI is both faster to train, consumes much less memory, and performs better.
>
> **W5: There should have been more challenging datasets in the experiments.**
> While we certainly agree that more challenging datasets exist, we chose these ones for a number of reasons. The datasets are all commonly used in the literature, which makes it easier to find competing methods and compare our work to others. The simpler datasets, such as rotated MNIST, are easy to understand intuitively and can therefore provide us with insights into the models' behaviours and their abilities to learn inductive biases. However, if you have specific datasets in mind, we will happily try to test LIVI on them and report back here. We should say, however, given compute limitations, for any dataset, the models should be able to run on a single GPU.
>
> **Q1: How does LIVI compare to standard MC sampling?**
> We compare the LIVI bounds to an importance sampling estimate in figure F.1 in the supplementary. Although the two bounds systematically underestimate the entropy (which makes sense as they are lower bounds), crucially, they largely follow the behaviour of the sampling estimate. This gives us some confidence that they are useful objectives. Note, however, that the experiment is only possible to do for very small models as importance sampling is challenging to get to work in more than a few hundred dimensions.
>
> **Q2: How does LIVI compare to other methods in terms of memory and runtime?**
> We agree that these are very important metrics and the only reason the runtime was not reported in the submission is that, during the deadline rush, the experiments were performed on different GPU devices and hence we did not have one-to-one time comparisons for all the methods. We have now completed benchmark measurements on MNIST, see table 4 in the attached PDF, which shows the training time required for convergence as well as the memory consumption. Compared to other expressive VI methods like AVB and MNF, LIVI is faster to train and consumes less memory.

---

> > ### Comment · Reviewer_M1uk · 2023-08-14
> > **Response to Authors**
> >
> > I thank the authors for the response. I also appreciate the effort in evaluating MNF on the UCI datasets toward this rebuttal. The main two shortcomings of this paper, in my opinion, remain.
> > * **Scalability issues**. First, I didn't mention transformers, even standard ResNet architectures generally have tens of millions of parameters which is more than one order of magnitude compared to the largest network used in this paper. Second, in terms of "relevant to the Bayesian deep learning community" - I respect the author's opinion, but I do not feel the same. I think that in this era of deep learning, an important aspect of the model is the ability to scale it beyond the network sizes in this paper. While I do not expect the method to be readily adjusted to massive networks, scaling to network sizes such as ResNet-18 and ResNet-34 is a reasonable requirement. The authors suggested several alternatives for scaling their model which sound great. In my opinion, the submission is incomplete without showcasing that.
> > * **Experimental section**. I stand behind my original comment that neither the datasets nor the baseline methods are strong enough. Therefore, I cannot attribute much value to the empirical evaluation in this paper when comparing LIVI to the proposed baseline methods. The authors wanted me to suggest alternatives. Well, in my opinion, this is the job of the authors as the setups in the paper are quite ubiquitous in the literature. Nevertheless, here are some suggestions:
> >    * In terms of datasets, CIFAR-100 is already more challenging than CIFAR-10, fine-grained classification datasets such as CUB, Cars, and Pets are another alternative. All appeared in the Bayesian literature before, and there are many more of course.
> >   * In terms of baselines to state a few, SWA/SWAG family which often shows good performance [1], deep kernel learning and its recent follow-up works [2], VI in function spaces and follow-up works [3], infinite-deep NNs [4, 5], and partially stochastic BNNs [6, 7].
> > There are other methods that I may have missed. I do not expect the authors to compare LIVI to all of these methods, but to at least some of them, I do.
> >
> > Overall I would like to state that I do value the proposed approach and its merits. Nevertheless, I think that currently, this paper is not ready to be published at NeurIPS. Hence, I decided to raise the score to 4 and not further.
> >
> > [1] Maddox, W. J., Izmailov, P., Garipov, T., Vetrov, D. P., & Wilson, A. G. (2019). A simple baseline for bayesian uncertainty in deep learning. Advances in neural information processing systems, 32.
> > [2] Wilson, A. G., Hu, Z., Salakhutdinov, R., & Xing, E. P. (2016, May). Deep kernel learning. In Artificial intelligence and statistics (pp. 370-378). PMLR.
> > [3] Sun, S., Zhang, G., Shi, J., & Grosse, R. (2018, September). FUNCTIONAL VARIATIONAL BAYESIAN NEURAL NETWORKS. In International Conference on Learning Representations.
> > [4] Nazaret, A., & Blei, D. (2022, June). Variational inference for infinitely deep neural networks. In International Conference on Machine Learning (pp. 16447-16461). PMLR.
> > [5] Xu, W., Chen, R. T., Li, X., & Duvenaud, D. (2022, May). Infinitely deep bayesian neural networks with stochastic differential equations. In International Conference on Artificial Intelligence and Statistics (pp. 721-738). PMLR.
> > [6] Sharma, M., Farquhar, S., Nalisnick, E., & Rainforth, T. (2023, April). Do Bayesian Neural Networks Need To Be Fully Stochastic?. In International Conference on Artificial Intelligence and Statistics (pp. 7694-7722). PMLR.
> > [7] Daxberger, E., Nalisnick, E., Allingham, J. U., Antorán, J., & Hernández-Lobato, J. M. (2021, July). Bayesian deep learning via subnetwork inference. In International Conference on Machine Learning (pp. 2510-2521). PMLR.

---

> > > ### Author Response · Authors · 2023-08-21
> > >
> > > Thank you for the detailed reply and for taking the time to provide a set of references for us to consider. We address your concerns below.
> > >
> > > **Scalability issues**
> > > We have started training LIVI to do inference in a WideResNet(28,10) on CIFAR-100. This architecture contains 36.5M parameters and has been used in multiple works like [1, 6, 8]. Without much tuning of LIVI, it achieves an accuracy of $76.7\\%$ and an NLL of $0.617$, which can be compared to $77.68\\% \\pm 0.29\\%$ and $0.944 \pm 0.002$ reported by [6] for full-network VI. We will continue tuning LIVI and include these results in our paper, but we wish to highlight that the hypernetworks used for LIVI in all experiments, including this, only contain about twice as many parameters as the networks they are modelling the posterior over - the same amount of parameters required for mean-field VI. Thus, we hope this experiment shows that LIVI can scale to modern network sizes.
> > >
> > > **Experimental section**
> > > While we respect the reviewer's opinion, we wish to state that we based our setup on experiments and baselines from [5, 7, 9]. By following their setups, as other works do too, a reader can compare and place LIVI in the broader literature on the topic. Moreover, we compare to KIVI [10] and AVB [11], which both focus on implicit VI, and hence we believe our experimental evaluation is up-to-date and comprehensive.
> > >
> > > Regarding the suggested references, we briefly discuss them below. We understand that the list is not meant to be comprehensive, but we wish to clarify why we did not include them in the submission. In short, LIVI is an approximate inference method where the downstream tasks are merely an assessment of the quality of our approximation, not the goal itself. We have therefore focused our experiments on comparisons with other approximate inference methods, in particular implicit VI methods, not general methods for solving the downstream tasks.
> > >
> > > SWAG, proposed in [1], was used as a baseline in [9], which we compare LIVI to. [9] found SWAG difficult to tune and DEs to be a stronger baseline overall, which is the reason we chose DEs to compare with. We will clarify this in the paper. Deep kernel learning, proposed in [2], aims at making Gaussian process (GP) modelling more expressive. As the GP posterior is available in closed form, this work is not directly related to the task we are trying to solve. [3] considers BNN inference in function-space using process-based inference, not distribution-based inference, which LIVI handles, so it is a different line of research. [4] introduces an infinitely deep BNN and a VI scheme for this model, and [5] focuses on continuous-depth neural networks, where they use an SDE to implicitly parametrise the posterior over the infinitely many weights. While both are interesting, they do not present inference methods for general BNNs, which is what we are concerned with. Both [6] and [7] focus on partially-stochastic networks in contrast to fully-stochastic networks, which was our primary target with LIVI. [6] is mainly a discussion and comparison paper, arguing that networks do not need to be fully stochastic - an open discussion in the community to which LIVI adds new evidence. While [6] compares a few simple strategies for choosing subsets of weights, their aim is to compare these to fully stochastic networks only, not to suggest an optimal selection strategy. They do show results for a WideResNet(28,10) on CIFAR-100, and we will thus compare to their results. The partially stochastic method of [7] builds on linearised Laplace, which we now compare to using the same library from [9] that both [6] and [7] use. We also consider last-layer Laplace approximations, which is a partially stochastic method as well.
> > >
> > > We do appreciate the list of references, many of which are important to discuss in the context of LIVI. We will therefore add them to our related works section.
> > >
> > > References:
> > > [1] Maddox et al. (2019). A simple baseline for Bayesian uncertainty in deep learning. NeurIPS.
> > > [2] Wilson et al. (2016). Deep kernel learning. AISTATS.
> > > [3] Sun et al. (2018). Functional Variational Bayesian Neural Networks. ICLR.
> > > [4] Nazaret & Blei (2022). Variational inference for infinitely deep neural networks. ICML.
> > > [5] Xu et al. (2022). Infinitely deep Bayesian neural networks with stochastic differential equations. AISTATS.
> > > [6] Sharma et al. (2023). Do Bayesian Neural Networks Need To Be Fully Stochastic? AISTATS.
> > > [7] Daxberger et al. (2021). Bayesian deep learning via subnetwork inference. ICML.
> > > [8] Nado et al. (2021). Uncertainty Baselines: Benchmarks for Uncertainty & Robustness in Deep Learning. arXiv:2106.04015.
> > > [9] Daxberger et al. (2021). Laplace Redux - Effortless Bayesian Deep Learning. NeurIPS.
> > > [10] Shi et al. (2018). Kernel Implicit Variational Inference. ICLR.
> > > [11] Mescheder et al. (2017). Adversarial Variational Bayes: Unifying Variational Autoencoders and Generative Adversarial Networks. ICML.

---

### Official Review · Reviewer_Q4Sr · 2023-07-06

**Soundness:** 3 good
**Presentation:** 4 excellent
**Contribution:** 4 excellent
**Rating:** 8
**Confidence:** 4

**Summary:**

This work proposes one approach to implicit variational inference that avoids using adversarial training. This is achieved by first applying a small amount of Gaussian noise to the implicitly generated weights $\theta$ to ensure that they are equipped with a valid distribution (enabling valid divergences), and then the now valid ELBO is approximated through linearizing the entropy regularization term. A further lower bound to this approximation is also proposed that allows for better scaling in high-dimensional settings, which is common for applying VI to neural models.

**Strengths:**

I felt that the approach presented was well justified, easy to follow in terms of motivation and development, and enables much more complex settings for variational inference to be applied to (in a stable manner) without having to suffer from the typical limiting mean-field assumptions.

The experimental results were impressive as well, seemingly achieving much better uncertainty quantification than the other competing methods while still retaining quality predictive performance.

**Weaknesses:**

The paper states in the contributions (line 49) that it "derive[s] a novel lower bound for variational inference"; however, I believe this is not quite the case. It is my understanding that the "novel lower bound" being mentioned here refers to $\mathcal{L}'$ and the further lower bound $\mathcal{L}''$ used for better scaling. This statement is in direct contradiction to equation 18 that states:
$\log p(\mathcal{D}) \geq \mathcal{L}(\gamma) \approx \mathcal{L}'(\gamma) \geq \mathcal{L}''(\gamma)$. Strictly speaking, $\mathcal{L}'$ is not a lower bound, but rather an approximation to an actual lower bound $\mathcal{L}$. To be clear, I do not think this is bad by any means, it just needs to be communicated clearly.

Aside from this, I think the experiments can be bolstered with a few additional comparisons. Namely:
- The impact of using $\mathcal{L}''$ over $\mathcal{L}'$ when the latter is still eligible (i.e., when using smaller models such as the UCI dataset tasks).
- How much performance (both in and out of distribution) differs between your proposed method and HMC (or other MCMC methods). This again would need to be done in low-dimensional settings, but I believe it should be feasible for at least a smaller neural network or linear model on some of the UCI dataset tasks.

Lastly, one of the contributions listed cited a novel generator architecture (line 53); however, the details of this seem to be relegated to the appendix. In future revisions, I believe if you are going to cite this as one of the major contributions then the details should at least be partially included in the main paper. I understand this probably wasn't done due to space limitations, but it is worth considering in my opinion.

**Questions:**

I do not have any direct questions, see the weaknesses for my main comments to be addressed please.

**Limitations:**

The authors did discuss the limitations adequately. Negative societal impact is not directly applicable here in my opinion.

---

> ### Author Rebuttal · Authors · 2023-08-09
>
> Thanks for a careful review of our work. We are delighted that you find our exposition easy to follow. We address you individual questions and suggestions below.
>
> **W1: The contributions state a novel lower bound, however, it is actually an approximation to a lower bound.**
> Thank you for highlighting this. We agree that we should have been more clear that $\mathcal{L}'$ is an approximation to a lower bound. We will clarify this in the paper.
>
> **W2: An experiment on the impact of using $\mathcal{L}''$ over $\mathcal{L}'$ would be good.**
> Thank you for suggesting this experiment - it would indeed be informative. We now include experiments to compare our two bounds, $\mathcal{L}'$ and $\mathcal{L}''$ on the UCI datasets, see tables 1 and 2 of the attached PDF. The results show that models trained with either $\mathcal{L}'$ or $\mathcal{L}''$ perform quite similarly both in terms of test log-likelihood and test RMSE, giving empirical justification for the lower bound on $\log\det(J J^\top)$.
>
> **W3: An experiment on the performance difference between LIVI and HMC in low-dimensional settings would be good.**
> This is also a great suggestion. For a qualitative assessment, Figure 1 of the main paper shows such an experiment for a toy problem. Furthermore, we now include experiments comparing our two bounds, $\mathcal{L}'$ and $\mathcal{L}''$, and HMC on the UCI datasets in tables 1 and 2 of the attached PDF. The results show that the model trained with $\mathcal{L}'$ performs close to or as well as HMC, suggesting that the local linearisation does not harm the expressivity dramatically. A model trained with $\mathcal{L}''$ is not much worse.
>
> **W4: The novel generator architecture should be briefly discussed in the main paper.**
> Thank you for pointing this out, it is a very good point. We will add a short description of the architecture to the main paper.

---

> > ### Comment · Reviewer_Q4Sr · 2023-08-14
> > **Response to Authors**
> >
> > Thank you for answering my concerns and providing more experimental results. I am satisfied by the response and maintain my original score assuming promised changes are incorporated in the camera-ready version of the paper.

---

### Official Review · Reviewer_C434 · 2023-07-06

**Soundness:** 3 good
**Presentation:** 3 good
**Contribution:** 3 good
**Rating:** 8
**Confidence:** 3

**Summary:**

The paper studies the problem of approximating high-dimensional multi-modal posteriors through neural samplers specifying implicit distributions. While Bayesian methods promise a variety of benefits in terms of generalization and calibrated predictions, in practice they have seen limited success due to intractability of exact Bayesian approaches and the tradeoffs made in approximate Bayesian methods. Implicit Variational Inference provides an alternative to approximating exact Bayesian posteriors by maintaining distributions implicitly by transforming samples from simple distributions, allowing it to admit much richer distributions. Implicit Variational Inference typically requires some density ratio based adversarial objectives, which can fail on high-dimensional problems (e.g. parameters of neural networks). The authors note two major issues in existing approaches a) KL being ill-defined due to the implicit density lying on a low dimensional manifold b) intractability of the entropy of the implicit density and its gradients. To tackle these issues, the authors first introduce Gaussian noise to the output of the sampler making it a Gaussian DLVM resulting in a well-defined KL over the parameter space.  Next the authors approximate the generator with a local linearization resulting in a Gaussian approximation of the output density, and obtain easy to compute approximation of the differential entropy of the output density. This results in a novel approximation to the ELBO based on the entropy approximation, which is scalable to high dimensional parameters. The authors discuss two variants based on computing the whole Jacobian or using a differentiable lower bound on the the determinant, which trade-off compute and quality of the approximation. Finally, the authors evaluate the method on a variety of tasks including impressive results on fairly large BNNs (WideResNet).

**Strengths:**

* The paper studies the important problem of approximating expressive Bayesian posteriors on high-dimensional spaces. Due to the general applicability and promising results, the work is significant and relevant to the community.
* To the best of my knowledge the main contributions of the paper namely addressing the ill-defined KL, local linearization for the entropy and the LIVI bound on the ELBO, are all novel.
* Introducing Gaussian noise to the sampler to induce a Gaussian DLVM is a neat and simple way of fixing the KL with minimal additional restrictions to the model
* Similarly, the local linearization of the neural sampler is a nice idea to obtain a cheaper estimator for the entropy.
* The experimental results are quite impressive - in particular the results on the WideResNet. I also appreciate the authors including the code with their submission to aid reproducibility of the results.
* Overall the paper is well-written with a clear exposition of ideas and most relevant details covered.

**Weaknesses:**

* The paper proposes a local linearization to make the entropy computation tractable. However, what is not clear to me is how the local-linearization affects the expressivity of the posterior and general performance. The experiments indicate that the effect is not large, but these are still “relatively” simple tasks so a thorough study of this would be useful
* The empirical comparisons also do not consider alternative approximate Bayesian methods.
* Some recent work [1] proposes a closely related approach which might be worth discussing in the paper

[1] Posterior Refinement Improves Sample Efficiency in Bayesian Neural Networks. Kristiadi et al., NeurIPS 2022.


**Questions:**

* What do you think would be the critical challenges in applying the method to even larger modern networks, e.g. transformers?
* What is the challenge in implementing the KIVI baseline? (since that is not included in all the experiments as the authors note)
* The runtime of the method is not explicitly mentioned anywhere in the paper except for the remark relative to Deep Ensembles in the appendix. This is important information which should be included in the main paper.
* Minor typos: L117 “in the following” -> “in this section” L123 “aGaussian” -> “a Gaussian”

**Limitations:**

Despite the impressive results in the paper, there still remains a gap between the models studied in the paper and the size of models considered in practice.

---

> ### Author Rebuttal · Authors · 2023-08-09
>
> Thank you for the insightful review. We very much appreciate that you find our work significant and relevant to the community and that the manuscript is well-written. We answer your specific concerns below.
>
> **W1: How does the local linearisation affect the posterior in terms of expressivity and performance?**
> This is a great question. We linearise the neural sampler $g_\gamma(z)$, but only when it is used within the estimation the intractable entropy and its gradients (see Eqs. (9) and (12)). However, this could indeed lead to sub-optimality in how we train the implicit approximation and the quality of the resulting distribution. We hope to obtain a highly flexible implicit distribution, and as you point out, our results indicate that the obtained posterior approximation is at least as good as the one found by our competitors.
>
> To investigate this further, we have added a comparison of our method with HMC on the UCI datasets, see tables 1 and 2 in the attached PDF. The results show that the model trained with $\mathcal{L}'$ performs close to or as well as HMC, suggesting that the local linearisation does not harm the expressivity dramatically. If the reviewer has suggestions for other experiments assessing the expressivity of the posterior, we would be happy to hear them.
>
>
> **W2: The empirical comparisons do not consider alternative approximate Bayesian methods.**
> We agree that comparing our proposed method to other approximate Bayesian methods is crucial. In the paper, we consider approximate Bayesian methods like MFVI, Laplace and Adversarial Variational Bayes, which also falls into the category of implicit methods. We now also include experiments with two versions of full Laplace, see tables 1 and 2 in the attached PDF. Furthermore, we have added results for multiplicative normalising flows (MNF, Louizos and Welling, 2017, [1]) on the UCI datasets (tables 1 and 2) and MNIST (table 3). The MNF are difficult to get to converge, however. All these results have been added to our paper too. If you have specific baselines in mind that you think we are missing to make a comprehensive benchmark, we will happily take them into consideration.
>
> [1] Louizos and Welling, "Multiplicative normalizing flows for variational Bayesian neural networks", ICML 2017.
>
> **W3: Posterior refinement (Kristiadi et al., 2022) is closely related and should be discussed.**
> Thank you very much for pointing us to this work. It is indeed relevant and we will add it to the related works section of the paper. Posterior refinement works by using the Laplace approximation as a clever base distribution for a normalising flow, which is then optimised to model the posterior of BNN. This can work well given a sufficiently expressive flow, which, however, typically comes at a high computational cost, especially in high dimensions. This is why Kristiadi et al. (2022) focuses on last-layer approximations. In contrast, LIVI uses a neural sampler to implicitly represent the posterior of the BNN, which means we can model the full posterior in an expressive manner. While there are pros and cons of both methods, it is difficult to imagine posterior refinement being scaled to millions of dimensions as we do here.
>
>
> **Q1: What is needed to apply LIVI to large, modern networks?**
> The hypernetwork poses the biggest challenge, in our opinion, as this is the network that is supposed to efficiently parametrise the high-dimensional implicit posterior. We do think that there are still ways to go beyond what we have done here. One approach would be to consider independent hypernetworks for each layer of the BNN, which would reduce the size of the individual hypernetworks at the cost of losing correlations across layers of the BNN. This itself should help such a framework to increase its modelling capacity significantly. One could also consider efficient ways of putting priors over deep networks to curb dimensionality like [1] that proposes implicit BNNs that have priors over activations rather than weights and biases, thus reducing the dimensionality of latent variables, and [2] which also assumes priors over units in a neural network and models weights using these latent distributions.
>
> [1] Trinh et al., "Scalable Bayesian neural networks by layer-wise input augmentation", arXiv 2020, https://arxiv.org/abs/2010.13498
> [2] Karaletsos et al., "Probabilistic Meta-Representations Of Neural Networks", UDL workshop 2018, https://arxiv.org/abs/1810.00555
>
>
> **Q2: Why was KIVI not implemented?**
> The official code for KIVI is written in a probabilistic programming subpackage developed by researcher at Tsinghua University and used by researchers there. The package is built with TensorFlow 1 and hence is not easy to port. We tried to implement our own version, but we were not confident in our implementation and decided to not include results from this method when they were not available in the original paper.
>
> **Q3: The runtime of the method should be included in the paper.**
> We agree that this is very a important metric by today's standards and we will include this in the main paper if space allows, otherwise the supplementary. The only reason the runtime was not reported in the submission is that, during the deadline rush, the experiments were performed on different GPU devices and hence we did not have one-to-one time comparisons for all the methods. We have now completed benchmark measurements on MNIST, see table 4 in the attached PDF, which shows the training time required for convergence as well as the memory consumption for LIVI and four other methods. When compared with other expressive VI methods like AVB and MNF, LIVI is faster to train and consumes less memory.
>
> **Q4: Minor typos.**
> Thank you very much for pointing these out. They have been corrected.

---

> > ### Comment · Reviewer_C434 · 2023-08-14
> > **Response to rebuttal**
> >
> > Thanks for the response and apologies for the delayed response!
> >
> > > How does the local linearisation affect the posterior in terms of expressivity and performance?
> >
> > Thanks for the clarification and additional results! It is indeed a bit challenging to test claims about expressivity, but I appreciate the additional experiment. It indeed appears to be the case that the performance / expressivity are not impacted considerably.
> >
> > > The empirical comparisons do not consider alternative approximate Bayesian methods.
> >
> > Thanks for the additional results.
> >
> > > Posterior refinement (Kristiadi et al., 2022) is closely related and should be discussed.
> >
> > Thanks for the explanation. I think this does merit some additional experimental validation but at the very least I hope the authors include this in the paper.
> >
> > > What is needed to apply LIVI to large, modern networks?
> >
> > Thanks for sharing these insights. On the topic of hypernetworks I would also mention recent advances (e.g. [1]) on this topic.
> >
> > > The runtime of the method should be included in the paper.
> >
> > Thanks for these results, these are indeed quite impressive and it would be great to have these in the paper.
> >
> > I am satisfied by the author response and encourage the authors to make the relevant changes for the camera ready version. I will maintain my score.
> >
> > [1] Knyazev, B., Hwang, D., & Lacoste-Julien, S. (2023). Can We Scale Transformers to Predict Parameters of Diverse ImageNet Models?. arXiv preprint arXiv:2303.04143.

---

### Official Review · Reviewer_MstJ · 2023-07-09

**Soundness:** 3 good
**Presentation:** 3 good
**Contribution:** 3 good
**Rating:** 8
**Confidence:** 4

**Summary:**

## Post Rebuttal Update

I have engaged with the authors for the rebuttal, and found their responses informative, prompting me to increase my score from a 7 to an 8.

## Original Review

The paper addresses two issues that are common in implicit variational inference methods such as amortised neural samplers, namely -

1. The implicit density often lies on low dimensional manifolds making the KL infinite/ill-defined
2. The gradients and entropy of the implicit density are intractable

The paper addresses these issues by (i) adding Gaussian noise to the output of the neural sampler, making it continuous w.r.t $\theta$, and (ii) linearising the neural sampler, resulting in a Bayesian linear model approximation to the non-linear model, giving closed-form solutions for the distributions. On different benchmarks the authors show that these methods work well on a range of small scale experiments compared to other state-of-the-art, such as last-layer Laplace and deep ensembles.

**Strengths:**

The paper is well motivated, has clear mathematical development, and solves common issues in implicit variational inference models. I like the use of linearisation to make the entropy and gradients tractable by getting closed-form solutions. The experiments are clear, and the baselines are well-considered. The results are also quite impressive in this domain, as it is often really difficult to beat deep ensembles, and the method seems to consistently perform really well.

**Weaknesses:**

In general I think the paper is well-written, however I have one major criticism -

1. I'm not convinced that the current approximation to $\log(J J^T)$ has been well-motivated or properly ablated. It would be nice to see some ablations on smaller-scale problems where calculating the full log determinant is tractable, and comparing it to the approximation made using just the highest singular value. Or doing a sweep over adding subsets of singular values vs the highest. Or even, plotting the eigenspectrum of $J J^T$ to show that it is true that the highest singular value is often much more dominant than the others.
2. If the paper is using the full-rank Jacobian of the neural sampler in order to estimate the entropy, I think a fairer comparison to make would be against full Laplace, not last-layer Laplace, which should be possible for UCI datasets and MNIST at least, using the Laplace library the authors cite.

**Questions:**

I think there are some really interesting connections between the obtained linearised implicit variational model, and existing methods such as linearised Laplace and regular variational inference, and I would love for there to be a small discussion about these things potentially. For example, in linearised Laplace, you assume a fully Bayesian posterior over a NN parameterised by $\theta$, then linearise the model around the MAP estimate, resulting in a tractable Gaussian approximation to the posterior. However, this posterior is more rich compared to a mean-field variational approximation, because the covariance of the posterior is given by $J_{\theta}^T \Lambda J_{\theta} + \sigma^2 I$, where $\Lambda$ is the prior over weights, and $\sigma^2$ is the noise variance. This form of the posterior looks very close to what is obtained by implicit variational inference, where the neural sampler can be considered similar to the NN model in linearised Laplace. I would be really interested in seeing discussions about these potential connections if possible. In fact, optimising the marginal likelihood for a linearised Laplace model is akin to doing ELBO with a Gaussian approximation. I would be really interested in seeing these parallels.
2. Given that the authors are using the entirety of the Jacobian of their neural sampler, and comparing to only last-layer Laplace, where only the last layer is modelled probabilistically, I would be really interested in seeing if they can run experiments on full linearised Laplace, especially on the small UCI and MNIST datasets, where this should be tractable. In fact, there are methods that perform full probabilistic inference for linearised Laplace using samples from the posterior, such as in [1].
3. I am also really interested in seeing what approximations to the Jacobian are best for estimating log det (J J^T), such as last-layer only, or through samples using Hutchinson's estimate, or by using more than one singular value and ablating through what the optimal number of singular values to consider is.

[1] Antorán, Javier, et al. "Sampling-based inference for large linear models, with application to linearised Laplace." arXiv preprint arXiv:2210.04994 (2022).

**Limitations:**

N/A.

---

> ### Author Rebuttal · Authors · 2023-08-09
>
> Thank you for your careful evaluation of our paper. We are happy that you liked our proposed method and found the results impressive. We respond to your specific questions and concerns below.
>
> **W1: The lower bound on $\log\det(JJ^\top)$ has not been motivated sufficiently nor properly ablated.**
> The motivation for the bound on $\log\det(J J^\top)$ comes from Geng et al. (2021) [1]. To clarify a possible source of confusion, the bound uses the *smallest* eigenvalue, not the largest. Essentially, it simply states that the sum of all log singular values is larger than (or equal to) the smallest singular value times the number of dimensions. While one can view this as a trivial and potentially quite loose bound, the smallest singular value is efficiently computed using the LOBPCG algorithm, and efficiency is very important for the high-dimensional problems we are focusing on in this work.
>
> To empirically assess the effect of the bound, we now include experiments to compare our two bounds, $\mathcal{L}'$ and $\mathcal{L}''$, which demonstrate the effect of the bound on the log determinant. The results, which can be found in tables 1 and 2 of the attached PDF, show that models trained with either $\mathcal{L}'$ or $\mathcal{L}''$ perform quite similarly both in terms of test log-likelihood and test RMSE, giving empirical justification for the lower bound on $\log\det(J J^\top)$.
>
> [1] Geng et al., "Bounds all around: training energy-based models with bidirectional bounds", NeurIPS 2021.
>
>
> **W2: Full Laplace is a fairer comparison than last-layer Laplace.**
> This is a great suggestion. Full Laplace using a full-rank Hessian is not feasible, even on MNIST, due to the size of the BNN that we use in our experiments. However, we have added results for two versions of full Laplace to the paper, 1) a non-linearised full Laplace using a low-rank approximation to the Hessian, and 2) a linearised full Laplace using a KFAC factorisation of a Generalised Gauss-Newton approximation to the Hessian,  see table 3 and figures 1 and 2 in the attached PDF. The non-linearised version of full Laplace underfits due to the quite crude approximation to the posterior over the full network. The linearised version works slightly better in terms of ECE and NLL on the rotated-MNIST benchmark (figure 1) and in the OOD entropy-CDF test (figure 2). Note, however, that linearised Laplace is not directly comparable to our proposed model, see the discussions for Q1 and Q2 below.
>
>
>
> **Q1: It would be nice to see a discussion on the connections between LIVI and existing methods such as linearised Laplace and regular variational inference.**
> We would like to clarify that we do not linearise the posterior; we linearise the neural sampler $g_\gamma(z)$, but only when it is used within the estimation the intractable entropy and its gradients (see Eqs. (9) and (12)). The resulting posterior approximation is, therefore, still a highly flexible (non-linearised) implicit distribution. It is correct that the terms appearing in a Laplace posterior and our linearised approximation are similar, but it is hard to draw direct analogies between the two approximate Bayesian methods as they are fundamentally quite different. Whereas linearised Laplace linearises the BNN and the predictive function, we linearise the neural sampler/hyper-network within the variational distribution $q_\gamma(\theta)$, thus obtaining the approximation $\tilde{q}_z(\theta)$, but only when it is needed to approximate the entropy. This is also the reason for why we do not linearise our BNN for predictions, which is common practice with linearised Laplace. We acknowledge that the distinction was not presented clearly enough in the original manuscript and will update the paper accordingly.
>
>
> **Q2: Please run experiments on full linearised Laplace on the UCI datasets and MNIST.**
> Thank you for this suggestion. We have added results for full Laplace without linearisation (low-rank Hessian) and full Laplace with linearisation (KFAC factorisation of a GGN approximation) to the paper and the PDF attachment here, see table 3 and figures 1 and 2 for results on MNIST.  We tried our best to make the Laplace library work for UCI regression but could not due to errors thrown by the second-order backends used (BackPack and Asdl), we will keep working on this and report back soon.While these are both relevant baselines, we would like to emphasise that, because of the reasons noted above, our results cannot be compared one-to-one with linearised Laplace as that posterior measures the uncertainty of a generalised linear model approximation and not the actual BNN.
>
>
> **Q3: Are other approximations to the Jacobian are better for estimating $\log \det (JJ^\top)$?**
> This is a very valid question, although one that is perhaps best answered in future work given the scope of it. We did not consider a last-layer-only approximation, as we have focused on the conventional Bayesian treatment where all latent variables (weights and biases) of the respective BNNs in all experiments have been modelled probabilistically. Basing the bound on more singular values than just the smallest one is a good idea, as it might give us a tighter bound. However, a motivating factor for using just the smallest singular value is that this can be found in linear time using the LOBPCG algorithm and, as we show in tables 1 and 2 in the attached PDF, the bound empirically works well. Using Hutchinson's estimator is a good idea too, however, empirically this estimator provides an upper bound, not a lower bound; please see Geng et al. (2021) who compare the smallest singular value estimate against Hutchinson's estimator for $\log \det(J^TJ)$.

---

> > ### Comment · Reviewer_MstJ · 2023-08-19
> >
> > Thanks for the rebuttal, and for the clarifications. I see now that I had a fundamental misunderstanding in where the models are being linearized, and it is indeed not fair to compare this method to Linearized Laplace. However, since LLA is a high performing method in this domain anyways, I appreciate the authors running LLA as a comparison. Regarding the linearized LLA version, I'm curious why the authors need to run a KFAC Approximation to the GGN matrix? Is it not possible to invert the GGN matrix in closed form using a decently sized Gpu? It might not be, and this is merely a matter of curiosity, as the current results the authors added are quite satisfactory to me.
> >
> > Thanks for the pointers regarding JJ^T, and the comparisons of L' and L'' as well, these clear up any doubts I might have had.
> >
> > I'm happy to increase my score to an 8, and thank the authors for an engaging rebuttal.

---

### Author Rebuttal · Authors · 2023-08-09

We thank all the reviewers for carefully reading our paper and providing constructive feedback. We appreciate that the reviews found that “the paper is well motivated, has clear mathematical development, and solves common issues” (MstJ) and that “experimental results were impressive as well” (Q4Sr). We have replied to each of you individually and have further attached a PDF with additional results that were requested. We look forward to further discussions with you.

---

### Decision · Program_Chairs · 2023-09-21

**Decision:**

Accept (spotlight)

**Comment:**

This paper proposed a new implicit VI method for Bayesian neural network approximate posterior inference. The idea is to use a neural sampler for the q distribution. The main contribution of this paper is a new methodology that solves two issues of implicit VI: (1) using semi-implicit posterior to solve the ill-defined KL divergence issue, and (2) using linearisation to approximate the intractable entropy term for the q distribution. Experiments on UCI and MNIST scale data demonstrates improved results regarding comparisons with other implicit VI methods and a few other baselines.

Reviewers all agree that the proposed approach is highly novel and sound, they commend that the method provides significant advances in implicit VI literature. One reviewer raised the question regarding small scale experiments and mentioned that experiments in current Bayesian deep learning literature is at least ResNet-18 or -50 level.

Overall implicit VI with neural sampler has yet to scale to ResNets with >10M parameters which is a downside of this paradigm of methods. But within this paradigm the proposed approach has made solid progress in my opinion, and the approach of using linearisation for implicit distribution entropy estimation is quiet interesting. Therefore I would recommend acceptance for this paper and encourage the authors to consider larger scale experiments in future work.